# Identification of positive and negative regulators of antiviral RNA interference in *Arabidopsis thaliana*

Si Liu[1], Meijuan Chen [1], Ruidong Li[2], Wan-Xiang Li[1], Amit Gal-On[3], Zhenyu Jia [2✉] & Shou-Wei Ding [1✉]

Virus-host coevolution often drives virus immune escape. However, it remains unknown whether natural variations of plant virus resistance are enriched in genes of RNA interference (RNAi) pathway known to confer essential antiviral defense in plants. Here, we report two genome-wide association study screens to interrogate natural variation among wild-collected *Arabidopsis thaliana* accessions in quantitative resistance to the endemic cucumber mosaic virus (CMV). We demonstrate that the highest-ranked gene significantly associated with resistance from both screens acts to regulate antiviral RNAi in ecotype Columbia-0. One gene, corresponding to *Reduced Dormancy 5* (*RDO5*), enhances resistance by promoting amplification of the virus-derived small interfering RNAs (vsiRNAs). Interestingly, the second gene, designated *Antiviral RNAi Regulator 1* (*VIR1*), dampens antiviral RNAi so its genetic inactivation by CRISPR/Cas9 editing enhances both vsiRNA production and CMV resistance. Our findings identify positive and negative regulators of the antiviral RNAi defense that may play important roles in virus-host coevolution.

[1] Department of Microbiology & Plant Pathology, University of California, Riverside, CA, USA. [2] Department of Botany & Plant Sciences, University of California, Riverside, CA, USA. [3] Department of Plant Pathology and Weed Science, Volcani Center, Rishon LeZion 7528809, Israel. ✉email: arthur.jia@ucr.edu; shou-wei.ding@ucr.edu

Diverse antiviral defense mechanisms have been documented in higher plants[1–4]. Several lines of evidence show that the RNA interference (RNAi) pathway mediates an essential antiviral immunity mechanism in plants[1,2]. In antiviral RNAi, host cells process viral long dsRNA into small interfering RNAs (siRNAs) by a Dicer nuclease to trigger specific virus clearance in RNA-induced silencing complex (RISC) by an Argonaute (AGO) protein[1,2,5–7]. Both Dicer and AGO gene families appear to have expanded in plants for the control of virus infection. In the model plant species *Arabidopsis thaliana*, for example, 3 of the 4 Dicer-like genes (DCLs) and 7 of the 10 AGO genes participate in the RNAi-mediated antiviral immunity[1,2,8–17]. Moreover, among the 6 RNA-dependent RNA polymerase (RdRP) genes of *A. thaliana*, RdRP 1 (RDR1), RDR2 and RDR6 have all been shown to direct viral siRNA amplification[1,2,18,19]. Interestingly, the 19-member AGO family of rice plants includes a hormone-inducible AGO18 that promotes antiviral RNAi by enhancing the expression of AGO1, which is necessary for vsiRNA-RISC assembly[20,21]. Notably, viral suppressors of RNAi (VSRs) are essential virulence proteins of plant RNA and DNA viruses[22,23]. Studies on the origins and variability of VSR genes have shown that antiviral RNAi exerts selection pressure against plant virus genomes[24–27].

Viruses with an RNA genome possess extraordinary adaptive abilities because of their error-prone replication mechanisms. Less is known about the natural variation that wild host plants in ecosystems with little human intervention accumulate and select to inherit in their progeny in response to virus infection[1–4]. Recently, whole genome information has become available for 1135 natural inbred lines from worldwide wild-collected accessions of *A. thaliana*[28]. Cucumber mosaic virus (CMV) is a natural pathogen of *A. thaliana* with incidence reaching up to 80% in some localities[29]. CMV contains a tripartite positive-strand RNA genome coding for 5 proteins required for genome replication, virion assembly, in-planta movement and RNAi suppression[30]. VSR protein 2b is encoded by an 'out-of-frame' gene that overlaps the more ancient viral RdRP gene and is the most variable viral protein among CMV strains[31–33]. CMV is transmissible by mechanical contacts and many species of aphids as well as less efficiently through the seed[30].

In this work, we investigate whether single nucleotide polymorphisms (SNPs) significantly associated with virus resistance among wild plant populations are enriched in specific pathways known to confer antiviral protection in plants. We use genome-wide association studies (GWAS) approach[34] to interrogate natural variation among re-sequenced wild *A. thaliana* populations in quantitative resistance[35] to CMV. For a better survey of host natural variation in antiviral responses, we conduct independent GWAS screens with two distantly related CMV strains, one unmodified and the other rendered defective in counter-defense against antiviral RNAi. We show that the highest-ranked gene significantly associated with quantitative virus resistance identified from each of two independent GWAS screens functions in antiviral RNAi. Notably, we demonstrate opposing roles in antiviral RNAi for the two genes evolved in ecotype Columbia-0, indicating that these regulatory genes of antiviral RNAi defense may play important roles in virus-host coevolution.

## Results

**Mapping host natural variation identifies a quantitative virus resistance gene.** We first assessed natural variation among the fully re-sequenced *A. thaliana* accessions in quantitative resistance to CMV-Δ2b, a previously characterized mutant of the highly virulent subgroup I CMV strain Fny that is rendered susceptible to antiviral RNAi by introducing nucleotide substitutions to prevent VSR-2b expression[13,36]. Virus accumulation levels in the upper non-inoculated leaves were measured by enzyme-linked immunosorbent assay (ELISA) of the viral coat protein (CP) two weeks after mechanical inoculation of *A. thaliana* seedlings.

The virus titers in 496 accessions displayed a nearly normal distribution after log-transformation (Fig. 1a and Supplementary Data 1). Analyzing the data with the easyGWAS pipeline[34] found that none of the SNPs significantly associated with quantitative virus resistance mapped to any of the known antiviral RNAi genes. The most significantly associated SNP resided in a region of chromosome 4 that codes for the gene At4g11040 (Fig. 1b, c and Supplementary Fig. 1a), corresponding to the previously reported *Reduced Dormancy 5 (RDO5)/Delay of Germination 18 (DOG18)*[37–39]. We found that CMV-Δ2b titers were significantly different between the accessions classified as haplotypes T and G according to the SNP (Supplementary Fig. 1b). Northern blot analysis further verified that CMV-Δ2b replicated to lower levels in 4 selected accessions of haplotype T including Col-0 than the 3 accessions of haplotype G (Supplementary Fig. 1c). We then compared CMV-Δ2b infection in accession Antwerpen-1 (An-1) carrying a single base-pair (bp) frameshifting deletion[39] in *RDO5* with accession Columbia-0 (Col-0), classified in the resistant haplotype along with 380 additional accessions among those examined (Fig. 1c and Supplementary Fig. 1b). Western blot analysis of the viral CP showed that CMV-Δ2b accumulated to much higher levels in An-1 plants than either Col-0 plants or the two independent lines of An-1 transgenically complemented with the *RDO5* gene from Col-0 (Fig. 1d). These results suggest suppression of CMV-Δ2b accumulation by *RDO5* from Col-0 plants.

To verify the role of *RDO5* in Col-0 plants, we obtained two Col-0 mutants carrying a T-DNA insertion at different positions in the second exon of *RDO5*, designated *rdo5-4* and *rdo5-5* (Fig. 1c). We also generated two independent transgene complementation lines of *rdo5-4* with the same *RDO5* gene driven by its native promoter described above. ELISA detection of the viral CP revealed significantly enhanced accumulation of CMV-Δ2b in both *rdo5-4* and *rdo5-5* mutant plants compared to either wild-type Col-0 plants or either of the two *RDO5*-complemented lines of *rdo5-4* plants (Fig. 1e). Moreover, both Western blotting detection of CP and Northern blotting detection of the viral genomic RNAs showed that CMV-Δ2b accumulated to higher levels in *rdo5-4* and *rdo5-5* mutant plants than wild-type Col-0 plants and the complemented lines of *rdo5-4* mutant (Fig. 1e, f). By comparison, CMV-Δ2b replicated to higher levels in *rdr6* mutant plants than *rdo5-4* and *rdo5-5* mutant plants (Fig. 1e, f), which may explain the absence of clear symptomatic differences between the infected Col-0 and *rdo5* mutant plants (Supplementary Fig. 2a).

To further verify the role of *RDO5* by an independent approach, we performed gene knockout via CRISPR/Cas9 in Col-0 plants and obtained another homozygous *rdo5* mutant containing a deletion of 344 bp starting from the 12th codon of *RDO5*, designated *cr15* (Fig. 1c and Supplementary Fig. 2b). We found that *cr15* plants also supported significantly enhanced replication of CMV-Δ2b compared to its wild-type Col-0 and the two transgene complemented lines of *rdo5-4* mutant (Fig. 1e, f). Together, our findings show that the highest-ranked gene in the GWAS screen confers quantitative virus resistance in Col-0 accession.

**RDO5 enhances antiviral RNAi by promoting viral siRNA amplification.** The seed-specific *RDO5* codes for a nuclear pseudophosphatase that promotes seed dormancy in a manner independent of the biosynthesis of phytohormone abscisic acid[37–39]. Consistently, *rdo5-4*, *rdo5-5* and *cr15* mutants exhibited

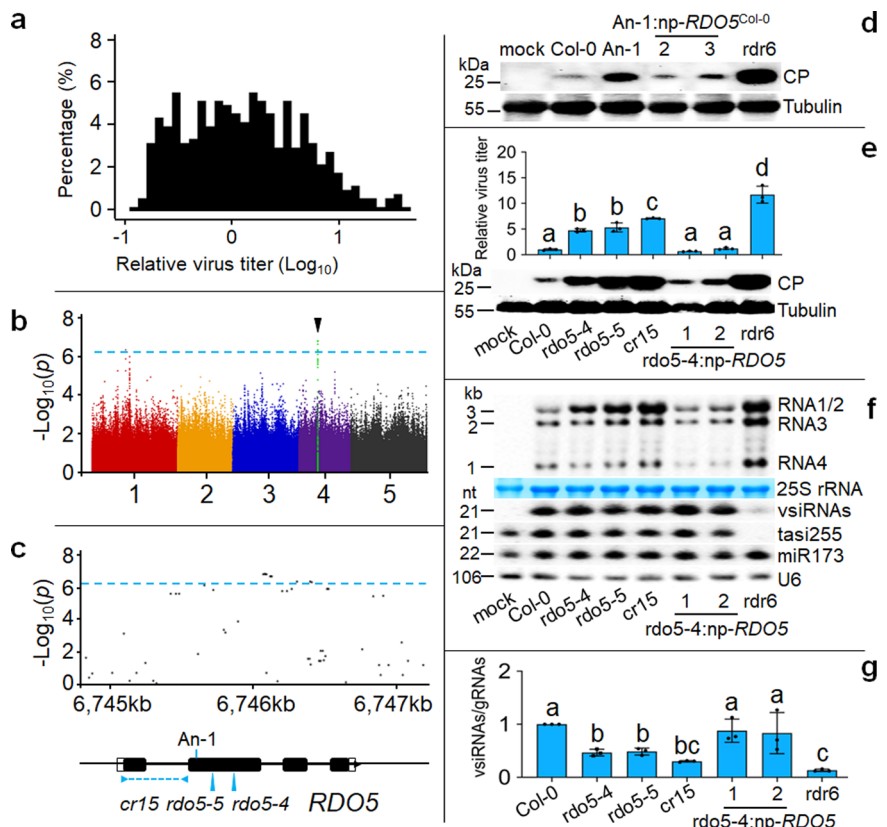

**Fig. 1 RDO5 confers quantitative virus resistance. a** A nearly normal distribution of CMV-Δ2b accumulation levels in 496 Arabidopsis accessions after ELISA readings at 2 weeks post-infection were log-transformed. **b** Manhattan plot for GWAS mapping of the quantitative resistance phenotype to CMV-Δ2b infection. Five *A. thaliana* chromosomes were depicted in different colors. The horizontal dash-dot line corresponds to the significance threshold ($p = 5.48 \times 10^{-7}$). The black triangle above the threshold indicates the most significantly associated locus. **c** Regional Manhattan plot (from 6745 kb to 6747 kb), structure of *RDO5*, and positions of T-DNA insertion alleles (*rdo5-4* and *rdo5-5*), CRISPR/Cas9 deletion allele (*cr15*) and the single base pare deletion in accession An-1. **d** Western detection of viral coat protein (CP) accumulation in An-1 and two *RDO5*-complemented lines. Detection of tubulin alpha chain was shown as loading control. **e, f** ELISA and Western detection (**e**) of viral CP accumulation with the titer in Col-0 set as 1 and Northern detection (**f**) of CMV-Δ2b genomic RNAs 1-3 (gRNAs), subgenomic RNA4 and vsiRNAs as well as endogenous miRNA 173 (miR173) and trans-acting siRNA 255 (tasi255) in wild-type (Col-0), *rdo5* mutants and complemented lines. 25 S rRNA was stained and U6 RNA probed on the same membrane as loading controls. **g** Ratios of vsiRNAs/gRNAs were calculated from Phosphor-imager readings of Northern hybridization signals in (**f**) with the ratio in Col-0 set as 1. The experiments in (**d**) and (**f**) were repeated three times independently with similar results. Data presented are means ± SEM from three replicates (**e**) or independent experiments (**g**), letters indicate significant differences (one-way ANOVA, Duncan, $p < 0.05$) and black dots represent the individual values. The source data underlying blots in (**d**), (**e**) and (**f**), ELISA data in (**e**) and ratio data in (**g**) are provided as a Source Data file.

significantly enhanced seed germination rates than Col-0 and the complementation lines of *rdo5-4* and An-1 plants carrying the wild-type *RDO5* transgene derived from Col-0 (Supplementary Fig. 2c), providing evidence for functional rescue by an additional phenotype. Notably, we found that CMV-Δ2b infection induced expression of *RDO5* in leaves, suggesting a different function of *RDO5* in vegetative tissues in response to virus infection (Supplementary Fig. 2d).

*A. thaliana* resistance to CMV-Δ2b is mediated by antiviral RNAi dependent mainly on the DCL4 and RDR6 pathway[13,19,33]. To investigate the mechanism of RDO5, we first compared the accumulation levels of the vsiRNAs in the same panel of wild-type and mutant infected plants used above for determining viral accumulation by Northern blot analysis (Fig. 1f). We found that the vsiRNAs accumulated to similar levels in wild-type Col-0 plants and the three *rdo5* mutants (*rdo5-4*, *rdo5-5* and *cr15*) despite the fact that CMV-Δ2b replicated to significantly higher levels in the *rdo5* mutant plants (Fig. 1e, f) and were thus expected to generate more abundant vsiRNA precursors for Dicer processing, suggesting deficiency of the mutant plants in the vsiRNA biogenesis. No marked differences were found in the

accumulation of endogenous siRNA or microRNAs between wild-type and *rdo5* mutant plants with or without CMV-Δ2b infection (Fig. 1f and Supplementary Fig. 2e). In *A. thaliana* plants infected with 2b-deficient mutants of CMV, the vsiRNAs are amplified by the host RDR1 and/or RDR6 so that loss of vsiRNA amplification in mutant plants leads to decreased ratios of vsiRNAs and viral genomic RNAs (gRNAs) although viral gRNAs accumulate to higher levels compared to the wild-type plants[19,40]. Thus, we measured the ratios of vsiRNAs and viral gRNAs detected in the infected wild-type and mutant plants. As expected, loss of RDR6-dependent vsiRNA amplification in the control *rdr6* mutant plants was associated with ~7-fold decrease of the vsiRNAs/gRNAs ratio compared to the resistant Col-0 plants (Fig. 1g). We observed significantly decreased vsiRNAs/gRNAs ratios in *rdo5-4*, *rdo5-5* and *cr15* mutant plants than those in Col-0 plants and the two *RDO5*-complemented lines of *rdo5-4* plants (Fig. 1g), suggesting a role of *RDO5* in the amplification of vsiRNAs.

To further investigate the mechanism of *RDO5*, we generated homozygous double or triple mutant plants by genetic crosses of *rdo5-5* plants with *rdr1* and *rdr6* single or double mutants.

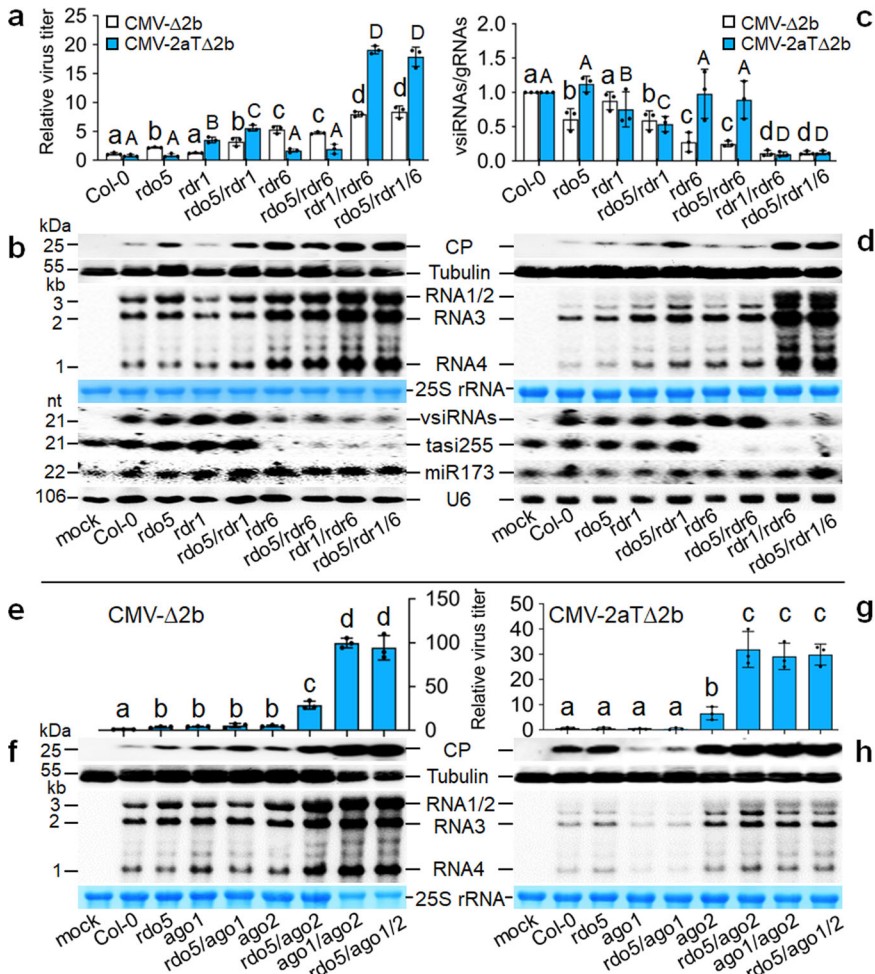

**Fig. 2 RDO5 promotes antiviral RNAi. a–h** Accumulation of CMV-Δ2b (**a**, **b**, **e**, **f**) or CMV-2aTΔ2b (**a**, **d**, **g**, **h**) detected in wild-type (Col-0), single, double or triple mutant plants as indicated at 2 weeks post-infection by ELISA (**a**, **e**, **g**) and Western blot analysis of the viral coat protein (CP) or Northern blot analysis of the viral RNAs 1-4 (**b**, **d**, **f**, **h**). Detection of the vsiRNAs and plant endogenous small RNAs (**b**, **d**), and the calculation of vsiRNAs/gRNAs ratios (**c**) were as described in the legend to Fig. 1. Note the reduced sample loading (to ½) for total proteins and total RNAs (**f**) from *ago1/2* double mutant plants and *rdo5/ago1/2* triple mutant plants infected with CMV-Δ2b. Data presented are means ± SEM from three replicates (**a**, **e**, **g**) or independent experiments (**c**), letters indicate groups with significant differences (one-way ANOVA, Duncan, $p < 0.05$) and black dots represent the individual values. The experiments in (**b**), (**d**), (**f**) and (**h**) were repeated three times independently with similar results. The source data underlying blots in (**b**), (**d**), (**f**) and (**h**), ELISA data in (**a**), (**e**) and (**g**), and ratio data in (**c**) are provided as a Source Data file.

Consistent with a predominantly RDR6-dependent antiviral RNAi against CMV-Δ2b described previously[13,36], CMV-Δ2b replicated to significantly lower levels in both wild-type Col-0 and *rdr1* plants than *rdr6* or *rdr1 rdr6* plants (Fig. 2a, b) and vsiRNA amplification in *rdr1* plants was as efficient as in Col-0 plants in contrast to defective vsiRNA amplification in *rdr6* or *rdr1 rdr6* plants (Fig. 2b, c). We found that CMV-Δ2b accumulation was significantly enhanced and the vsiRNAs/gRNAs ratio was significantly decreased in *rdo5 rdr1* double mutant plants compared to *rdr1* single mutant plants (Fig. 2a–c). By contrast, no significant differences in either CMV-Δ2b accumulation or the vsiRNAs/gRNAs ratio were observed between *rdr6* and *rdo5 rdr6* plants (Fig. 2a–c). Moreover, neither CMV-Δ2b accumulation nor the vsiRNAs/gRNAs ratio was significantly different between *rdr1 rdr6* double mutant plants and *rdo5 rdr1 rdr6* triple mutant plants (Fig. 2a–c). These findings indicate that *RDO5* acts specifically in the antiviral RNAi defense mechanism by enhancing vsiRNA amplification in an RDR6-dependent pathway.

We performed an additional set of infection experiments to verify the proposed RDR6-dependent antiviral activity of *RDO5*

using CMV-2aTΔ2b, which contains a 295-nt deletion in RNA 2 of Fny-CMV resulting in both the loss of VSR-2b expression and a C-terminal truncation of the viral RdRP protein[19]. Unlike CMV-Δ2b, efficient CMV-2aTΔ2b infection occurs only in *rdr1 rdr6* double mutant plants because it triggers potent amplification of vsiRNAs by both RDR1 and RDR6 pathways[13,19]. ELISA and Western blotting detection of the viral CP as well as Northern blotting detection of the viral RNAs found no significant differences in the accumulation of CMV-2aTΔ2b and the vsiRNAs/gRNAs ratios between either wild-type plants (Col-0) or *rdr6* mutant plants with *rdo5* mutant plants (*rdo5-4, rdo5-5* and *cr15*) or the two complemented lines of *rdo5-4* plants (Supplementary Fig. 3). These results indicate that *RDO5* is dispensable for RDR1-dependent amplification of the vsiRNAs. Consistently, enhanced accumulation of CMV-2aTΔ2b and decreased vsiRNAs/gRNAs ratio were observed in *rdo5 rdr1* plants compared to *rdr1* plants, but no significant differences in either CMV-2aTΔ2b accumulation or the vsiRNAs/gRNAs ratio were observed between *rdr6* and *rdo5 rdr6* plants (Fig. 2a, c, d). Moreover, we detected no significant differences in either CMV-2aTΔ2b accumulation or the vsiRNAs/gRNAs ratio between *rdr1*

*rdr6* plants and *rdo5 rdr1 rdr6* plants (Fig. 2a, c, d). These findings show that *RDO5* confers virus resistance in the antiviral RNAi pathway dependent on RDR6, but independent of RDR1.

It is known that AGO1 and AGO2 act cooperatively to direct antiviral RNAi against CMV-Δ2b by associating with two distinct sets of vsiRNAs predominantly with 5′-terminal U and A, respectively[13]. We generated an additional set of homozygous double or triple mutants by combining *rdo5-5* with *ago1-27* (hypomorphic allele) and/or *ago2-1* (null allele). We found that CMV-Δ2b replicated to similar levels in *ago1 ago2* and *rdo5 ago1 ago2* plants (Fig. 2e, f). Similarly, no significant differences in virus accumulation were detected between the double and triple mutant plants after infection with CMV-2aTΔ2b (Fig. 2g, h). Thus, *RDO5* exhibited no antiviral activity against either CMV-Δ2b or CMV-2aTΔ2b when both AGO1 and AGO2 were not functional, demonstrating that the antiviral activity of *RDO5* is completely dependent on AGO1 and AGO2. Interestingly, whereas both CMV-Δ2b and CMV-2aTΔ2b replicated to significantly higher levels in *rdo5 ago2* plants than *ago2* plants (Fig. 2e–h), combining *rdo5-5* with *ago1-27* did not enhance the accumulation of neither CMV-Δ2b nor CMV-2aTΔ2b (Fig. 2e–h), suggesting that *RDO5* was involved in the *AGO1*, but not the *AGO2* pathway.

Together, our mechanistic studies show that identified by GWAS mapping as the highest-ranked gene with natural variation significantly associated with resistance, *RDO5* confers virus resistance in Col-0 plants in the antiviral RNAi pathway by promoting RDR6-dependent vsiRNA amplification.

**Natural variation identifies a host gene that inhibits antiviral defense.** In a parallel GWAS screen, we used a wild-type isolate of subgroup II CMV strain Q (Q-CMV) that is not modified in counter-defense against antiviral RNAi, but causes much weaker disease symptoms than Fny-CMV in Col-0 plants. We measured the accumulation levels of Q-CMV in 500 accessions of *A. thaliana* by detecting the viral genomic RNA3 with quantitative reverse transcription-polymerase chain reaction (RT-qPCR) (Supplementary Fig. 4a and Supplementary Data 1). Using the easyGWAS pipeline[34], we did not identify SNPs significantly associated with quantitative resistance in any of the known antiviral RNAi genes as found in the above GWAS screen. The SNP associated most significantly with quantitative resistance against Q-CMV resided in a region of chromosome 5 between genes At5g05130 and At5g05140 (Fig. 3a, b and Supplementary Fig. 4b), neither of which was characterized previously. We found that Q-CMV accumulation levels as measured by RT-qPCR were significantly different between the accessions classified as haplotypes A and G according to the SNP (Supplementary Fig. 4c). Northern blot analysis further verified that Q-CMV replicated to lower levels in 3 selected accessions of haplotype G than the 4 accessions of haplotype A including Col-0 (Supplementary Fig. 4d).

Functional studies in Col-0 plants identified At5g05140 as a regulator of antiviral RNAi, designated *Antiviral RNAi Regulator 1* (*VIR1*). Firstly, we found that expression of *VIR1*, but not At5g05130, was induced in Col-0 plants by infection with Q-CMV, Q-CMV-Δ2b or CMV-2aTΔ2b (Supplementary Fig. 5a). *VIR1* expression levels were also significantly higher in the selected accessions of haplotype A than those from haplotype G after Q-CMV infection (Supplementary Fig. 5b). As shown in Fig. 3b, we obtained *VIR1* and At5g05130 knockout mutants of Col-0 plants. *vir1-1* mutant contained a T-DNA insertion in the eighth exon of *VIR1*. *cr5* mutant was generated by CRISPR/Cas9 genome editing to delete a 215-bp fragment in the first exon of At5g05130. Neither *vir1-1* nor *cr5* exhibited visible developmental defects (Supplementary Fig. 6a). RT-qPCR analysis showed that

Q-CMV accumulated to significantly lower levels in *vir1-1* mutant plants than Col-0 (Fig. 3c). In contrast, no statistically significant differences in Q-CMV accumulation were found between Col-0 and At5g05130-*cr5* mutant plants or two independent *VIR1*-transgene complementation lines of *vir1-1* (Fig. 3c). Moreover, *vir1-1* plants, but not *cr5* plants or the transgene-complemented lines of *vir1-1*, supported significantly reduced replication of Q-CMV-Δ2b (Fig. 3c), a 2b-deletion mutant of Q-CMV defective in suppressing antiviral RNAi initiated by either DCL2 or DCL4 shown by a previous study[33]. We next calculated and compared the ratio of the vsiRNAs/ gRNAs detected by Northern blotting (Fig. 3d–f). The results indicated that production of the vsiRNAs triggered by either Q-CMV or Q-CMV-Δ2b was significantly enhanced in *vir1-1* mutant plants compared to Col-0 plants, *cr5* mutant plants or the two *VIR1*-complemented lines of *vir1-1* (Fig. 3d–f).

To further verify the role of *VIR1*, we generated two additional *vir1* mutants by CRISPR/Cas9 genome editing, designated *cr6-14* and *cr6-31*, which respectively contained an insertion and deletion of single cytosine nucleotide (C) in the 15th codon of *VIR1* (Supplementary Fig. 6b). Similar to *vir1-1*, both *cr6-14* and *cr6-31* mutants exhibited enhanced resistance to either Q-CMV or Q-CMV-Δ2b since both viruses accumulated to lower levels and triggered production of more abundant vsiRNAs in *cr6-14* and *cr6-31* mutants than Col-0 plants (Fig. 3c–f). However, we observed no obvious differences in the accumulation of endogenous siRNA and miRNAs between wild-type and *vir1* mutant plants either with or without CMV infection (Fig. 3d, f and Supplementary Fig. 6c). These findings indicate that the highest-ranked gene in the independent GWAS screen dampens antiviral RNAi in Col-0 accession against both Q-CMV and Q-CMV-Δ2b by inhibiting production of the vsiRNAs.

**VIR1 is a negative regulator of antiviral RNAi.** The N-terminal region of VIR1 protein shares strong homology with Arabidopsis transcript elongation factor IIS (AtTFIIS) conserved broadly in eukaryotes (Supplementary Fig. 5c). However, most of the critical residues in the C-terminal domain of TFIIS essential to facilitate mRNA synthesis by RNA polymerase II complex are not conserved in VIR1 protein[41]. Similar to *rdo5* mutant plants, *rdo2* mutant plants that lack AtTFIIS display clearly reduced seed dormancy[42]. We found that *vir1-1* mutant plants exhibited significantly enhanced seed dormancy and that *VIR1* expression was detectable in seeds (Supplementary Fig. 6d, e), suggesting *VIR1* as a negative regulator of seed dormancy.

To determine whether VIR1 interferes with vsiRNA biogenesis, we constructed double and triple mutants by genetic crosses of *vir1-1* plants with *dcl2-1* and/or *dcl4-2* mutant plants characterized previously[33]. We challenged this panel of mutant plants with CMV-2aTΔ2b and CMV-Δ2b derived from Fny-CMV since our previous studies have defined the genetic pathways in the biogenesis of vsiRNAs triggered by these mutant viruses[13,19]. RT-qPCR and Northern blotting analysis showed that CMV-2aTΔ2b replicated to significantly lower levels in *vir1-1* mutant plants than Col-0 plants (Fig. 4a, b), as did Q-CMV and Q-CMV-Δ2b (Fig. 3c–f). Notably, whereas no significant differences were detected in virus accumulation between *dcl2* and *vir1 dcl2* mutant plants, CMV-2aTΔ2b replicated to significantly higher levels in *vir1 dcl4* double mutant plants than *dcl4* single mutant plants (Fig. 4a, b). Consistent with the known dominant role of DCL4 over DCL2 in the biogenesis of vsiRNAs[9–11,19,33], 21-nt vsiRNAs made by DCL4 were undetectable whereas 22-nt vsiRNAs by DCL2 accumulated to high levels in either *dcl4* plants or *vir1 dcl4* plants (Fig. 4b). By comparison, however, 22-nt vsiRNAs accumulated to markedly lower levels in *vir1 dcl4* plants than

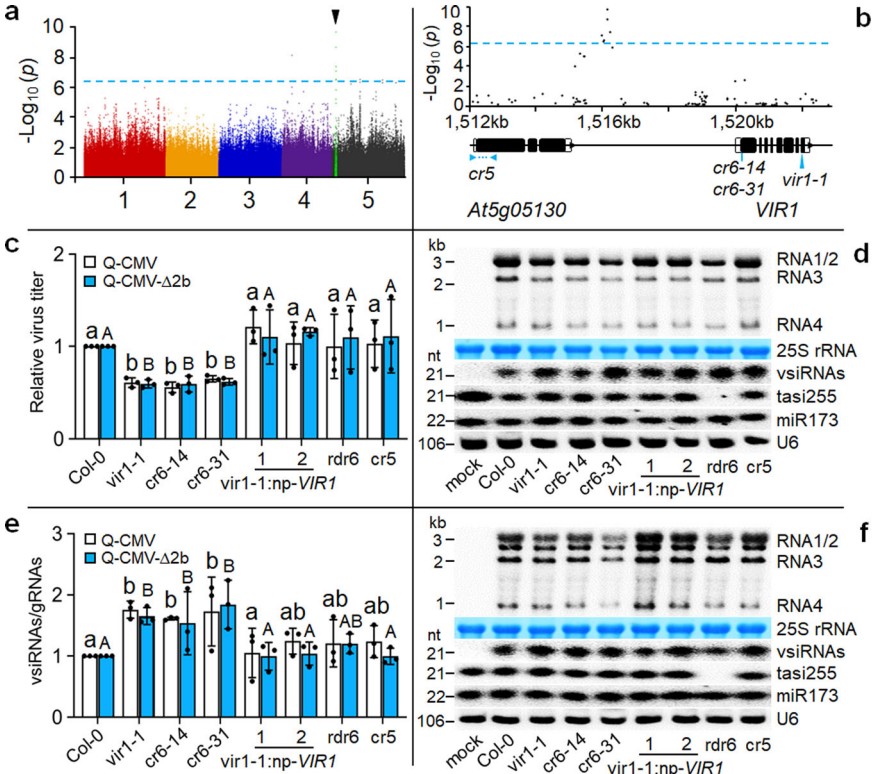

**Fig. 3 Identification of *VIR1*. a** Manhattan plot for GWAS mapping of the susceptibility phenotype to Q-CMV infection. The horizontal dash-dot line corresponds to the significance threshold ($p = 5.45 \times 10^{-7}$). The black tringle above the threshold indicates the most significantly associated locus. **b** Regional Manhattan plot (from 1512 kb to 1522 kb), structure of *VIR1*, and positions of T-DNA insertion allele *vir1-1* and CRISPR/Cas9 genome edited alleles (*cr5* in At5g05130 and *cr6-14* and *cr6-31* in *VIR1*). **c, d, f,** Accumulation of Q-CMV (**c, d**) or Q-CMV-Δ2b (**c, f**) detected in wild-type (Col-0), *cr5* mutant, *vir1* mutants (*vir1-1*, *cr6-14* and *cr6-31*), or two independent *VIR1*-complemented lines of *vir1-1* at 2 weeks post-infection by RT-qPCR analysis of the viral RNA3 (**c**) or Northern blotting of the viral RNAs 1-4 (**d, f**). The vsiRNAs and plant endogenous small RNAs were also detected by Northern blotting (**d, f**). **e** The vsiRNAs/gRNAs ratios were calculated as described in the legend to Fig. 1. Data in (**c**) and (**e**) are means ± SEM from three independent experiments, letters indicate groups with significant differences (one-way ANOVA, Duncan, $p < 0.05$) and black dots represent the individual values. The source data underlying blots in (**d**) and (**f**), qRT-PCR data in (**c**) and ratio data in (**e**) are provided as a Source Data file.

*dcl4* plants (Fig. 4b), indicating that activation of DCL2-dependent antiviral RNAi by 22-nt vsiRNAs in the absence of DCL4 is upregulated by *VIR1*.

RT-qPCR analysis revealed significant suppression of *DCL4* induction by CMV-2aTΔ2b in Col-0 plants compared to *vir1-1* mutant plants (Fig. 4e). By contrast, *DCL2* expression was induced by CMV-2aTΔ2b in *dcl4* single mutant plants, but not in *vir1 dcl4* double mutant plants or any other wild-type and mutant plants examined (Fig. 4e). Reduced induction of *DCL4* in Col-0 plants and strong induction of *DCL2* in *dcl4* mutant plants were both observed after infection with Q-CMV (Supplementary Fig. 7). These findings together suggest a model in which *VIR1* negatively regulates antiviral RNAi in wild-type plants by restricting transcriptional induction of *DCL4*, but upregulates 22-nt vsiRNA-directed antiviral RNAi by transcriptional induction of *DCL2* in the absence of DCL4.

We observed no statistically significant differences in CMV-2aTΔ2b accumulation between *dcl2 dcl4* and *vir1 dcl2 dcl4* mutant plants (Fig. 4a, b). This result indicates that although *VIR1* enhanced antiviral defense in *dcl4* single mutant, it failed to do so in *dcl2 dcl4* double mutant, consistent with the proposed role of *DCL2* in the upregulation of antiviral RNAi by *VIR1*. As reported previously[33], neither of the double and triple mutant plants produced 21- or 22-nt vsiRNAs. The results from CMV-Δ2b infection in the same panel of mutants (Fig. 4c, d) essentially reproduced those from CMV-2aTΔ2b infection (Fig. 4a, b). For example, genetic inactivation of *VIR1* enhanced the resistance to

CMV-Δ2b in Col-0 plants, but promoted virus susceptibility and reduced DCL2-dependent production of 22-nt vsiRNAs in *dcl4* plants (Fig. 4c, d). However, CMV-Δ2b replicated to the same levels in *dcl2 dcl4* and *vir1 dcl2 dcl4* mutant plants (Fig. 4c, d). Together, our findings indicate that in addition to a role in seed dormancy, *VIR1* negatively regulates antiviral RNAi in Col-0 plants by reducing vsiRNA production possibly by restricting transcriptional induction of *DCL4*.

## Discussion

In this work, we searched for the natural variation among wild *A. thaliana* populations that is most significantly associated with quantitative resistance to an endemic RNA virus. We conducted independent GWAS screens with wild-type Q-CMV and the VSR-deficient CMV-Δ2b, respectively. Unlike Fny-CMV, from which CMV-Δ2b was derived, Q-CMV replicates to higher levels in *dcl2 dcl4* mutant plants than wild-type Col-0 plants[33,36], indicating incomplete suppression of antiviral RNAi by Q-CMV. Surprisingly, none of the SNPs significantly associated with quantitative virus resistance from both of our GWAS screens and an additional GWAS screen reported recently by others[43] mapped to any of the known antiviral RNAi genes, including AGO2 shown recently to exhibit natural variation in non-host virus resistance[44]. However, genetic studies show that both of the highest-ranked gene significantly associated with quantitative virus resistance identified from each of our GWAS screens

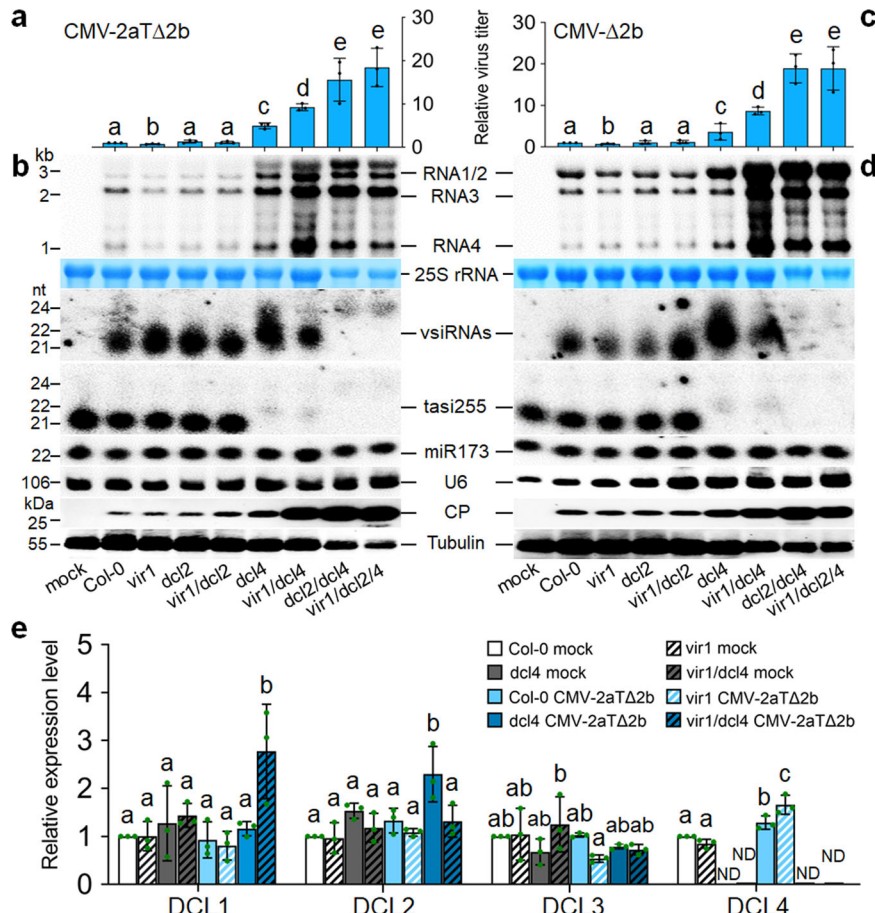

**Fig. 4 VIR1 is a negative regulator of antiviral RNAi. a–d** Accumulation of CMV-2aTΔ2b (**a**, **b**) or CMV-Δ2b (**c**, **d**) detected in wild-type (Col-0), single, double or triple mutant plants as indicated at 2 weeks post-infection by RT-qPCR analysis of the viral RNA3 (**a**, **c**), Northern blotting of the viral RNAs 1–4 (**b**, **d**), or Western blotting of the viral CP (**b**, **d**). The vsiRNAs and plant endogenous small RNAs were also detected by Northern blotting (**b**, **d**). Note the reduced sample loading (to ½) for total proteins and total RNAs from *dcl2/4* double mutant plants and *vir1/dcl2/4* triple mutant plants (**b**, **d**). **e** RT-qPCR analysis of *DCL1*, *DCL2*, *DCL3* and *DCL4* mRNA levels in wild-type (Col-0), single or double mutant plants without infection or one week post-infection with CMV-2aTΔ2b as indicated. *A. thaliana EF1α* (*At5g60390*) mRNA was used as internal control. Data in (**a**), (**c**) and (**e**) are means ± SEM from three independent experiments, letters indicate groups with significant differences (one-way ANOVA, Duncan, $p < 0.05$) and green dots represent the individual values. ND not determined. The source data underlying blots in (**b**) and (**d**), and qRT-PCR data in (**a**), (**c**) and (**e**) are provided as a Source Data file.

function in antiviral RNAi. Further studies on the two identified genes evolved in Columbia-0 accession demonstrate opposing roles in antiviral RNAi. Our findings provide direct evidence to support antiviral RNAi as a dominant defense mechanism in virus-host coevolution, which is consistent with previous genetic studies that identify antiviral RNAi as an essential antiviral defense in plants[1,2,5,6,8–17,20,21].

Most of the antiviral RNAi pathway genes characterized to date have been identified by their known activity in experimentally induced RNAi to target mRNAs transcribed in the nucleus[2,7,8,18,19,45–49]. Recently, mutant *Arabidopsis thaliana* and *Caenorhabditis elegans* defective in antiviral RNAi have been isolated by sensitized genetic screens using viruses or viral RNA replicons rendered inactive in RNAi suppression[36,50–52]. In this work, we showed that antiviral RNAi pathway genes can be identified by GWAS mapping of natural variation in quantitative virus resistance among wild-collected *A. thaliana* accessions. Our findings illustrated the technical feasibility of using GWAS mapping for unbiased identification of host genes critical for virus resistance. Given the increasing availability of genomic sequences for crops[53,54], this approach may allow for identification of key plant genes essential for regulating crop resistance to important viral pathogens.

Whereas *VIR1* has not been characterized before this work, *RDO5* has a known function to promote seed dormancy with the mechanism yet to be defined. Our mechanistic studies demonstrate that *RDO5* specifically enhances antiviral RNAi by promoting amplification of the vsiRNAs in a pathway dependent on RDR6, but independent of RDR1. By contrast, *VIR1* dampens antiviral RNAi by restricting production of the vsiRNAs and suppresses seed dormancy in wild-type Col-0 plants. Our results suggest that VIR1 may act by blocking viral induction of *DCL4* in a manner similar to the dominant negative mutant version of AtTFIIS shown recently to modify the transcriptome in *tfIIs* mutant plants[41]. Interestingly, *VIR1* is necessary for the transcriptional induction of *DCL2* and the upregulation of *DCL2*-dependent antiviral RNAi by 22-nt vsiRNAs in the absence of DCL4, which may explain at least in part the dominant role of DCL4 over DCL2 in the biogenesis of vsiRNAs known since 2006[9–11,19,33]. However, it is unknown whether *VIR1* modulates the functional roles of *DCL2* that are active in the presence or absence of DCL4 in uninfected plants[9,55–58]. Notably, we show that *VIR1* inactivation by CRISPR/Cas9 genome editing confers resistance to CMV either active or defective in RNAi suppression, providing a strategy to generate transgene-free virus resistant plants. In summary, our findings indicate that *RDO5* and *VIR1*

have opposing roles in both antiviral RNAi and seed dormancy. We propose that a shared mechanism is under natural selection to regulate antiviral RNAi and seed dormancy.

## Methods

**Viruses and plant materials.** Mutant viruses CMV-Δ2b and CMV-2aTΔ2b were derived from the subgroup I strain Fny-CMV isolated and cloned in New York from a muskmelon farm[59]. In CMV-Δ2b, three AUG codons at the first (start codon), 8th, and 18th positions of 2b ORF encoded by wild-type RNA2 of Fny-CMV were mutated to ACG so that the amino acids encoded in the +1 overlapping 2a ORF were not altered[13]. CMV-2aTΔ2b contained a 295-nt deletion in the 2b coding sequence, which also removed the C-terminal 80 amino acids of the viral RdRP 2a protein[13,19,60]. Q-CMV is a subgroup II strain isolated in Australia and molecularly cloned in 1995 following passages since 1964 in laboratory host plants[61]. In Q-CMV-Δ2b, the 2b coding sequence of RNA2 was deleted and replaced with CCCGGG, which also removed the C-terminal 68 amino acids of the 2a protein[7,33]. All viruses were purified after propagation in *Nicotiana clevelandii*. Virion concentration used for mechanical inoculation of *A. thaliana* was 10 μg/mL for both CMV-Δ2b and Q-CMV and 20 μg/mL for both CMV-2aTΔ2b and Q-CMV-Δ2b.

*A. thaliana* natural accessions of CS78942, T-DNA insertion mutants *rdo5-4* (SALK_206727), *rdo5-5* (SALK_120098) and *vir1-1* (SALK_020870) were obtained from the Arabidopsis Biological Resource Center (https://abrc.osu.edu). Single and double mutants in *DCL*, *RDR* and *AGO* families were as described[9–11,13,19,33,49,62,63]. Additional double and triple mutants of *rdo5-4* or *vir1-1* were generated by genetic crosses. Seeds were vernalized at 4 °C in dark for 3 days and seedlings grown in growth room with 10 h light - 14 h dark cycle at 23 °C.

*RDO5*, *VIR1* and At5g05130 knockout mutants were generated in Col-0 background by CRISPR/Cas9 genome editing as described[64] with the sequences of guide RNAs (gRNAs) listed in Supplementary Data 2. Briefly, transgenic seeds were screened on MS plates containing 20 μg/mL hygromycin B (Invitrogen) and T1 plants grown in soil for genotyping by PCR and sequencing with primer pairs listed in Supplementary Data 2. T1 plants with DNA editing in the target gene were backcrossed with Col-0 to identify mutants homozygous for the edited allele without Cas9 transgene. The *RDO5* and *VIR1* genomic fragments (gRDO5 and gVIR1) including 2 kb native promotor region upstream of the start codon, were PCR amplified from Col-0 plants by Phusion DNA polymerase (NEB) with the primer pairs listed in Supplementary Data 2 and verified by sequencing after cloning into pENTR/D-Topo. The resulting pENTR/D-gRDO5 and pENTR/D-gVIR1 were then transferred to PGWB616 by Gateway LR rection enzyme mix (Invitrogen), yielding PGWB616-gRDO5 and PGWB616-gVIR1, respectively. These constructs were used in transgene complementation for *RDO5* in An-1 and *rdo5-4* plants and for *VIR1* in *vir1-1* plants by *Agrobacterium*-mediated plant transformation using the floral dipping method[65]. Transgenic plants were selected with 0.1% basta spraying and self-pollinated to generated two independent lines for each transgene. T3 seedlings were used for virus inoculation.

**Genome-wide association studies mapping.** Seedlings of 496 and 500 *A. thaliana* accessions (Supplementary Data 1) from the 1001 genome project[28] were infected by mechanical inoculation with CMV-Δ2b and Q-CMV, respectively. The upper systematically infected leaves from 16 plants inoculated per accession were harvested at two weeks post-inoculation and pooled for viral accumulation measurements in three technical replicates. The accumulation level of CMV-Δ2b was determined by ELISA as described[66] using the mouse monoclonal antibody (1:2000 dilution) specific to the coat protein (CP) of the CMV subgroup I strain provided kindly by Dr. Xueping Zhou. Briefly, 0.3 g pooled leaves freshly harvested from each accession were ground in 1 mL 0.05 M NaHCO₃ (pH 9.6) with cocktails protease inhibitor (Complete). After centrifugation, the supernatant was diluted with sample loading buffer and used for ELISA detection in triplicate with the CP antibody, alkaline phosphatase-conjugated secondary antibody (Invitrogen, WP20006, 1:2000 dilution), and P-nitrophenyl phosphate (Thermo) using an infinite 200 Pro plate reader (Tecan i-control). The supernatant from mock-inoculated Col-0 plants was included as negative control. Virus titer in each sample was calculated according to the standard curve generated using a serial dilution of CMV-Δ2b virions (0, 0.1, 0.5, 1, 5, 10 and 20 μg/mL). The accumulation level of Q-CMV was determined by RT-qPCR detection in triplicate of the viral RNA3 in the total RNA extracted from the pooled leaves of each accession. 1 μg total RNA was used for reverse transcription (RT) and cDNAs diluted 10 times for subsequent quantitative PCR analysis by CFX96 Real-Time System with SYBR Green Supermix (Bio-Rad). mRNA of *EF1α* gene (At5g60390) was used as internal reference as reported previously[67]. Primer pairs for the detection of Q-CMV RNA3 and *EF1α* mRNA were listed in Supplementary Data 2.

Association mapping was conducted based on the 1001 genomes data[28] using the easyGWAS pipeline[34] (https://easygwas.ethz.ch/), TAIR10 gene annotations and the EMMAX algorithms[68]. The minor allele frequency cutoff was set at 5% such that 1,823,394 and 1,833,408 SNPs were used for GWAS mapping of CMV-Δ2b and Q-CMV data from 496 and 500 accessions, respectively. Based on the calculation using the ANOVA approach, the heritability was 0.9727 for the CMV-Δ2b screen trait and 0.9059 for the Q-CMV screen trait, respectively. The pairwise Pearson correlations calculated between the three replicates of virus detection were

0.976, 0.970 and 0.972 in CMV-Δ2b screen, and 0.909, 0.905 and 0.925 in Q-CMV screen. The mapping trait was the mean value of three replicates of the qPCR on the viral RNA3 for Q-CMV screen and the log-transformed mean value of three replicates of ELISA on the viral CP for CMV-Δ2b screen. We applied a $p$ value threshold of $1/m$ to GWAS mapping, where $m$ is the number of SNPs used for the analysis[69–72]. The R program "qqman" was used to create Manhattan plots and quantile-quantile plots based on the results of the GWAS mapping. Two-sample $t$ test was performed to compare virus titers of different haplotypes using GraphPad Prism 7, with $p = 0.05$ as the significance level.

**Characterization of virus infection.** To characterize virus infection, upper systematically infected leaves from 16 plants were harvested at two weeks post-inoculation and pooled for total RNA and protein extraction[73]. Five and 20 μg total RNAs were loaded each lane for Northern detection of the viral genomic and vsiRNAs by α-³²P- and γ-³²P-labelled probes, respectively[73]. Hybridization signals were detected by phosphor imager Typhoon 9410 and analyzed by ImageQuant TL 7.0 (GE Healthcare). Ratios of vsiRNAs vs viral genomic RNAs (gRNAs) were calculated from Phosphor-imager readings of Northern hybridization signals[19,40]. With the primer pairs listed in Supplementary Data 2 and *EF1α* mRNA as the internal reference[67], RT-qPCR was used to detect the accumulation of *RDO5*, *VIR1*, *DCL1*, *DCL2*, *DCL3* or *DCL4* mRNA in upper non-inoculated leaves of plants one week after mock or virus inoculation. For Western blotting, total proteins were separated on 12% polyacrylamide gel before transferred to 0.45 μm nitrocellulose membrane (GVS North America). Monoclonal antibody (1: 5000 dilution) specific to Fny-CMV CP[66] was used for viral CP detection and probing with *A. thalian* tubulin alpha chain specific antibody (Agrisera, AS20 4483, 1:5000 dilution) as loading control, Goat anti-mouse IgG (H + L) (Invitrogen, G-21040, 1:2000 dilution) was used as secondary antibody. All experiments were biologically repeated at least three times. We examined the expression pattern of *VIR1* in *A. thaliana* from the Arabidopsis RNA-Seq Database (http://ipf.sustech.edu.cn/pub/athrna/)[74] using 960 libraries generated from Col-0.

**Seed dormancy assay.** For seed dormancy assays[37], 50 newly harvested seeds were sown on wet filter paper in 9 cm diameter culture dishes, and incubated in growth room with 16 h light - 8 h dark cycle at 23 °C. Seed germination rates for each of the genotypes were determined 7 days after incubation with 4 independent repeats.

**Reporting summary.** Further information on research design is available in the Nature Research Reporting Summary linked to this article.

## Data availability

Data supporting the findings of this work are available within the paper and its Supplementary Information files. A reporting summary for this Article is available as a Supplementary Information file. Source data are provided with this paper.

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

## Acknowledgements

We wish to thank Drs. Daoxin Xie and Xueping Zhou for sharing materials. This project was supported by funding from the Agricultural Experimental Station and College of Natural and Agricultural Sciences, the University of California, Riverside (to S.-W.D.) and by grants from the US-Israel Binational Agricultural Research and Development Fund (no. IS-5027-17C to A.G.O. and S.-W.D.) and Office of Sponsored Research of the King Abdullah University of Science and Technology, Saudi Arabia (no. OSR-2015-CRG4-2647 to Drs. Magdy Mahfouz and S.-W.D.).

## Author contributions

S.L. designed and performed experiments, analyzed data, and wrote the paper. S.-W.D. conceived, designed, and supervised the study, and wrote the paper. Z.J. designed and supervised the GWAS screens and statistical analysis, and wrote the paper. M.C., R.L., W.-X.L. performed experiments or analyzed data. A.G.-O. provided intellectual input to the study. All authors revised and provided feedback for the final version of the paper.

## Competing interests

The authors declare no competing interests.
