## [Peer Review File · Nature Communications]

Identification positive and negative regulators of antiviral RNA interference in *Arabidopsis thaliana*Reviewers' Comments:

Reviewer #1:

Remarks to the Author:

In this very interesting study two *Arabidopsis thaliana* genes involved in quantitative resistance/susceptibility to cucumber mosaic virus (CMV) are identified in two independent GWAS. The genes are identified as modifiers of RNAi antiviral defence by means of genetic and functional analyses. The authors conclude that antiviral RNAi is a main factor in plant adaptation to virus infection. Conclusions are based on well performed experiments and sound results, and are important and of general interest for the public of Nature Communications.

However, I think the paper would benefit if the following comments were considered:

1. Title. Even if, in my opinion, results sustain that antiviral RNA is a major driver of plant adaptation to virus infection, and even if GWAS was performed in collections of wild genotypes, there is no proof that antiviral RNAi drives adaptation in natural ecosystems. We do not know if results of GWAS would have been the same had they been performed under field conditions, for instance. Thus, I suggest deleting "in natural ecosystems" from the title. This would not at all diminish the interest of the study, or its attraction to potential readers.
2. The first GWAS was performed with a subgroup I CMV strain impaired in VSR-2b expression, while the second one used a subgroup II wild type virus. The first GWAS allowed identifying a gene that enhances resistance and the second one a gene that dampens resistance. The authors should explain briefly the rationale for using such different inoculum for each GWAS, and the relationship with the results.
3. Line 59. Indicate here that the virus used is Fny-CMV-delta2b, so that the reader does not need to go to the supplement to get this information.
4. Lines 117-124. A sentence explaining why the ratio vsRNA/gRNA gives more significant information than virus titre should be added.
5. Line 263. In relation with comment 1, I do not suggest modifying this sentence, which is cautious enough.

Reviewer #2:

Remarks to the Author:

This report essentially describes two different studies aimed at identifying host factors involved in quantitative resistance to viruses in plants. The authors have undertaken two GWAS studies to identify genes displaying natural variation responsible for differing susceptibility to CMV in a large number of *Arabidopsis* ecotypes and describe the identification of two genes, AVI3 and AVI4.

The AVI3 story is relatively straightforward in that they identify naturally occurring KO alleles and test KO alleles in Col-0 to demonstrate that a lack of functional AVI3 results in increased susceptibility to a VSR-compromised CMV. To understand how AVI3 might function, they generate a number of crosses with lines mutated in RDR and AGO genes. From the results, they conclude that AVI3 enhances vsRNA amplification. This conclusion is largely based on quantification of virus and vsRNAs in double/triple mutants. The results from the genetic analyses are consistent with the conclusions. However, without more direct data it is unclear how AVI3, a putative phosphatase family member, functions in this process.

The second story in this manuscript is the identification of a second locus, AVI4, using a strain of CMV that is naturally less virulent, to undertake the same kind of GWAS analysis. The basis for this gene underlying genetic variation in virus resistance is less clear (see major issue below) and the authors do not follow up on this aspect once identifying AVI4. Furthermore, the effect on susceptibility is moderate. As such, the same genetic analysis approach used with AVI3 may not be informative because the other RNAi components have so much greater effect than AVI4.

Overall, the data is well executed and controlled and the writing is clear. However, there is a lack of converging experimentation to demonstrate that AVI3 and AVI4 are directly involved in RNAi-mediated antiviral defense. AVI3 is putative phosphatase that appears to be involved in germination,

whereas AVI4 is annotated as a transcription elongation factor. Indeed, the effects of AVI4 may be due to alterations in the transcription (albeit very moderate) levels of DCL genes rather than being directly involved in RNAi. As such, although these genes are certainly of interest, their direct mechanistic involvement in RNAi remains to be elucidated.

Major issue:

The basis of the natural variation of AVI4 is not clear. The SNP identified is intergenic. The authors then study KO alleles and overexpressing lines to demonstrate the role of AVI4 in virus resistance. However, they do not demonstrate/discuss what are the differences between different ecotypes. For example, is AVI4 induced more or less in response to virus in different ecotypes? Are there functionally different SNPs?

If the authors wish to convince the reader that natural variation in AVI4 is relevant to virus resistance, it is necessary to demonstrate the effects of this natural variation in addition the KO analyses.

Minor issues:

Line 38 : "Interestingly, the hormone-inducible AGO18 of the rice 19-member AGO family..." This sentence is confusing.

Line 41: "Studies on the origins and variability of viral suppressors of RNAi (VSR) genes have shown that antiviral RNAi exerts selection pressure against plant virus genomes⁴⁻⁸." The first four references cited describe what VSRs are and how they work. Only reference 8 refers to studies pertaining to selection pressure on virus genomes.

Line 247: typo in AVI4

Depending on the sequencing projects of from which the data was generated for different ecotypes, the degree of coverage is not always the same and sometimes somewhat low. Was the frameshift mutation in At4g11040 validated for the An-1 ecotype and as well as a number of other ecotypes with altered susceptibility relative to Col-0? How common is this specific mutation? The text implies that most of the ecotypes have the same allele as Col-0, but whether the rest are identical to An-1 is not directly stated.

The gene At4g11040 is listed as DOG18 or RDO5 in the Arabidopsis database. Does this gene really need another name?

The authors state that "Thus, AIV4 became inactive to interfere with antiviral defense when neither DCL2 nor DCL4 was functional."

However, this statement may go a bit far. Given the very strong effects of deleting DCL2 and DCL4 versus the relatively moderate effect of deleting AVI4, it seems more likely that any effects of AVI4 deletion are simply masked by the more important effects of the DCLs.

The discussion is very brief and there are some issues that could use more discussion including the following:

In the discussion the authors begin with "We show in this work that the natural variation evolved in association with quantitative virus resistance in wild *A. thaliana* populations is under the strongest selection for an optimal antiviral RNAi response. Our findings provide direct evidence to support antiviral RNAi as a dominant defense mechanism to drive host genetic adaptation to viral infection in natural ecosystems."

Although this statement is intuitively in line with general thinking, this study does not demonstrate this per se. This study does not demonstrate that natural variation evolved in association with

quantitative resistance. In fact, given that these genes are probably involved in myriad other processes (ie. Germination, transcriptional termination) the association could be pleiotropic and should be discussed accordingly. Furthermore, this one study does not demonstrate that RNAi is a dominant defense mechanism. Many other studies have hinted at this in the past.

The authors suggest that AVI3 and AVI4 are important factors in determining host-virus interactions. However, the screens have been done with viruses that are extremely compromised in virulence. As such, any conclusions about natural host-virus interactions regarding these genes needs to be tempered in this light.

Reviewer #3:

Remarks to the Author:

This is a nice tidy paper and the presented data mostly support the conclusions that AVI3 and AVI4 are regulators of the RDR6 and DCL2 antiviral pathways respectively. The results will be of interest to those in the field, they are an advance on the current literature. Methodology is sound.

I have no substantive comments about the experimental GWAS approach. I do have reservations, however, about the evolutionary interpretation that natural variation in the two genes is due to adaptation of the antiviral response. The two AVI genes may well have more general and pleiotropic effects and the authors should not rule out that the variation is associated with adaptive change in these other phenotypes. AVI3, for example, is a seed dormancy co-factor. The analysis in this paper does not rule out that the polymorphism in AVI3 is associated with dormancy or indeed with any other, as yet unidentified, phenotypes of this gene. At the very least the authors should mention the possibility that these genes might be associated with trans-kingdom RNAi associated with fungal and oomycete pathogens even if they cannot test it for this paper.

It would be helpful to have a bit more information about the GWAS alleles of AVI4. The interpretation is that the encoded protein is a negative regulator of antiviral RNAi but the GWAS alleles and the cr5 CRISPR allele all have more quantitative resistance to the CMV isolates – the opposite of what would be expected. If that is the case the GWAS mutations would have affected negative regulatory elements in the AVI4 promoter so that the encoded protein would be more abundantly expressed in the accessions with the GWAS variant alleles.

One interpretation of AVI4 is based on the observation that DCLs antagonise DCL21. If AVI4 promotes this antagonism it would explain why 22nt vsiRNA was less abundant in *avi4/dcl4* than in DCL4 (Fig 4b and d). In tomato at least there are endogenous miRNAs and siRNAs that are dependent on SIDCL2. It would be interesting to know what happens to the Arabidopsis equivalent in *avi4*.

1. Bouché, N., Laressergues, D., Gascioli, V., Vaucheret, H. & Bouche, N. An antagonistic function for Arabidopsis DCL2 in development and a new function for DCL4 in generating viral siRNAs. *EMBO J.* 25, 3347–56 (2006).

Reviewer #4:

Remarks to the Author:

According to the communications with the editors, I would mainly comment on the GWAS methodologies and the corresponding statistical analysis performed in this study. GWAS was performed by the EMMAX algorithm implemented in the pipeline "easyGWAS". The EMMAX algorithm is a widely used GWAS algorithm based on the linear mixed model, which is the state-of-the-art method for GWAS in structured populations. Generally, I didn't detect any major drawback in the statistical analysis of GWAS. But the following points are of my concern:

1) Before GWAS, we would always need to check, if possible, the heritability of the trait (or more precisely, the heritability of the phenotypic values that serve as the input of the GWAS model) because this information tells us the proportion of genetic variation which can be exploited in GWAS. In the Methods section it was mentioned that "all experiments were repeated at least three times" (lines 578-579) and the mapping trait was the mean values of the virus level (measured by ELISA or qPCR, lines 556-558). I would like to ask how large were the correlations between different replicates. Or how was the repeatability of the "phenotypic data". Strictly speaking, one needs data from several environments/independent experiments to estimate the heritability of the data, which is difficult for some traits like the virus levels inspected in this study. Then, the genomic heritability using linear mixed models may be an alternative. Have the authors considered this?

2) The authors did not mention how the missing values in the marker data were handled. Did you filter the markers with a certain threshold of missing rate? Did you impute the (remaining) missing values before GWAS?

3) The threshold of P values chosen by the authors could be a problem in the statistical point of view. It was mentioned that a threshold of $1/m$ was applied in GWAS, where m is the number of SNPs used in the analysis (line 558-559). This threshold is far too liberal if one considers the family-wise error rate, for which the threshold is usually $0.05/m$ (the Bonferroni correction). I certainly understand that the Bonferroni correction is in many cases too stringent. But as far as I know, there are three other standard methods implemented in easyGWAS (Benjamini-Hochberg, Benjamini-Yekutieli and Storey-Tibshirani). I am wondering why the authors did not apply these methods. I see that the authors cited a review paper (Zhang et al. 2019) for the choice of the threshold $1/m$. However, this threshold was described as a "subjective and less stringent method" in the paper, which means that there is no solid statistical theory supporting the choice of this liberal threshold. Of course, experimental verification was done in this study to confirm the function of the candidate genes identified in GWAS, making the choice of threshold a less essential point. Nevertheless, it would be better if one could avoid using subjective thresholds.

Point-by-point response to the reviewers' comments

Reviewer #1

In this very interesting study two *Arabidopsis thaliana* genes involved in quantitative resistance/susceptibility to cucumber mosaic virus (CMV) are identified in two independent GWAS. The genes are identified as modifiers of RNAi antiviral defence by means of genetic and functional analyses. The authors conclude that antiviral RNAi is a main factor in plant adaptation to virus infection. Conclusion are based on well performed experiments and sound results, and are important and of general interest for the public of Nature Communications.

However, I think the paper would benefit if the following comments were considered:

1. Title. Even if, in my opinion, results sustain that antiviral RNA is a major driver of plant adaptation to virus infection, and even if GWAS was performed in collections of wild genotypes, there is no proof that antiviral RNAi drives adaptation in natural ecosystems. We do not know if results of GWAS would have been the same had they been performed under field conditions, for instance. Thus, I suggest deleting “in natural ecosystems” from the title. This would not at all diminish the interest of the study, or its attraction to potential readers.

Response: As suggested, “in natural ecosystems” has been deleted from the title in the revised manuscript. We agree that although the *Arabidopsis* accessions used in this work were derived from wild-collect individuals in different geographical locations, we have evaluated their response to virus infection under the same set of growth room conditions to avoid introducing many environmental variables.

2. The first GWAS was performed with a subgroup I CMV strain impaired in VSR-2b expression, while the second one used a subgroup II wild type virus. The first GWAS allowed identifying a gene that enhances resistance and the second one a gene that dampens resistance. The authors should explain briefly the rationale for using such different inoculum for each GWAS, and the relationship with the results.

Response: We have added text to give more detail about the two viruses used for GWAS screens (see lines 68-71 and lines 195-197) and to explain the rationale “For a better survey of host natural variation in antiviral responses, we conducted independent GWAS screens with two distantly related CMV strains, one unmodified and the other rendered defective in counter-defense against antiviral RNAi” (see lines 57-60).

3. Line 59. Indicate here that the virus used is Fny-CMV-delta2b, so that the reader does not need to go to the supplement to get this information.

Response: Done - see lines 70-72.

4. Lines 117-124. A sentence explaining why the ratio vsRNA/gRNA gives more significant information than virus titre should be added.

Response: One sentence has been added: “In *A. thaliana* plants infected with 2b-deficient CMV mutants, the vsiRNAs are amplified by the host RDR1 and/or RDR6 so that loss of vsiRNAs amplification in mutant plants leads to decreased ratios of vsiRNAs and viral genomic RNAs (gRNAs) even though viral gRNAs accumulate to higher levels compared to the wild-type plants”. See lines 133-136.

5. Line 263. In relation with comment 1, I do not suggest modifying this sentence, which is cautious enough.

Response: This sentence is now located in lines 308-310: “Our findings provide direct evidence to support antiviral RNAi as a dominant defense mechanism to drive host genetic adaptation to viral infection in natural ecosystems”. Thanks!

Reviewer #2:

This report essentially describes two different studies aimed at identifying host factors involved in quantitative resistance to viruses in plants. The authors have undertaken two GWAS studies to identify genes displaying natural variation responsible for differing susceptibility to CMV in a large number of Arabidopsis ecotypes and describe the identification of two genes, AVI3 and AVI4.

The AVI3 story is relatively straightforward in that they identify naturally occurring KO alleles and test KO alleles in Col-0 to demonstrate that a lack of functional AVI3 results in increased susceptibility to a VSR-compromised CMV. To understand how AVI3 might function, they generate a number of crosses with lines mutated in RDR and AGO genes. From the results, they conclude that AVI3 enhances vsiRNA amplification. This conclusion is largely based on quantification of virus and vsiRNAs in double/triple mutants. The results from the genetic analyses are consistent with the conclusions. However, without more direct data it is unclear how AVI3, a putative phosphatase family member, functions in this process.

The second story in this manuscript is the identification of a second locus, AVI4, using a strain of CMV that is naturally less virulent, to undertake the same kind of GWAS analysis. The basis for this gene underlying genetic variation in virus resistance is less clear (see major issue below) and the authors do not follow up on this aspect once identifying AVI4. Furthermore, the effect on susceptibility is moderate. As such, the same genetic analysis approach used with AVI3 may not be informative because the other RNAi components have so much greater effect than AVI4.

Overall, the data is well executed and controlled and the writing is clear. However, there is a lack of converging experimentation to demonstrate that AVI3 and AVI4 are directly involved in RNAi-mediated antiviral defense. AVI3 is putative phosphatase that appears to be involved in germination, whereas AVI4 is annotated as a transcription elongation factor. Indeed, the effects of AVI4 may be due to alterations in the transcription (albeit very moderate) levels of DCL genes rather than being directly involved in RNAi. As such, although these genes are certainly of interest, their direct mechanistic involvement in RNAi remains to be elucidated.

Major issue:

The basis of the natural variation of *AVI4* is not clear. The SNP identified is intergenic. The authors then study KO alleles and overexpressing lines to demonstrate the role of *AVI4* in virus resistance. However, they do not demonstrate/discuss what are the differences between different ecotypes. For example, is *AVI4* induced more or less in response to virus in different ecotypes? Are there functionally different SNPs? If the authors wish to convince the reader that natural variation in *AVI4* is relevant to virus resistance, it is necessary to demonstrate the effects of this natural variation in addition the KO analyses.

Response: To address the major issue, we have added experiments and analysis in new panels of Supplementary Figs. 4 and 5 from previously completed experiments that were not included in the first submission, which had space restriction. We found that Q-CMV accumulation levels as measured by RT-qPCR were significantly different between the accessions classified as haplotypes A and G according to the identified intergenic SNP (Supplementary Fig. 4c). Northern blot analysis further verified that Q-CMV replicated to lower levels in 3 selected accessions of haplotype G than the 4 accessions of haplotype A including Col-0 (Supplementary Fig. 4d). Moreover, *AVI4* expression levels were significantly higher in the selected accessions of haplotype A than the selected haplotype G accessions after Q-CMV infection (Supplementary Fig. 5b), suggesting enhanced induction of *AVI4* in Col-0 and other haplotype A accessions that support higher Q-CMV replication compared to haplotype G accessions. However, we found no additional SNP within the *AVI4* locus that is significantly associated with Q-CMV titers among natural ecotypes (Fig. 3b). See lines 206-210 and lines 214-216.

Minor issues:

Line 38: “Interestingly, the hormone-inducible AGO18 of the rice 19-member AGO family...” This sentence is confusing.

Response: We have modified the sentence as “Interestingly, the 19-member AGO family of rice plants includes a hormone-inducible AGO18 to promote antiviral RNAi by enhancing the expression of AGO1 necessary for vsRNAs-RISC assembly.”

Line 41: “Studies on the origins and variability of viral suppressors of RNAi (VSR) genes have shown that antiviral RNAi exerts selection pressure against plant virus genomes⁴⁻⁸.” The first four references cited describe what VSRs are and how they work. Only reference 8 refers to studies pertaining to selection pressure on virus genomes.

Response: We have modified the ref list by adding ref. 27-29.

Line 247: typo in *AVI4*

Response: Fixed - thanks

Depending on the sequencing projects of from which the data was generated for different ecotypes, the degree of coverage is not always the same and sometimes somewhat low. Was the frameshift mutation in At4g11040 validated for the An-1 ecotype and as well as

a number of other ecotypes with altered susceptibility relative to Col-0? How common is this specific mutation? The text implies that most of the ecotypes have the same allele as Col-0, but whether the rest are identical to An-1 is not directly stated.

Response: The frameshift mutation in At4g11040 of ecotype An-1 has been verified by DNA sequencing by the same group of researchers who reported *Reduced Dormancy 5* (*RDO5*), but renamed the gene as *Delay of Germination 18* (*DOG18*) in their second paper (Xiang et al, 2016). They found that the single nucleotide deletion is extremely rare, present in only six of the 870 accessions examined. This allele was not specifically considered in our study because the standard GWAS mapping protocol used in our analysis considers only alleles with a frequency cutoff at 5%.

We have also added data from previously completed experiments that were not included in the first submission. Northern blot analysis verified that CMV- Δ 2b replicated to lower levels in 4 selected accessions of haplotype T including Col-0 than the 3 accessions of haplotype G (Supplementary Fig. 1c). See lines 83-85.

The gene At4g11040 is listed as *DOG18* or *RDO5* in the Arabidopsis database. Does this gene really need another name?

Response: We have added *DOG18* to this gene – thanks. As pointed out, this seed-specific gene was given two different names (*DOG18/RDO5*) by the same group of researchers in their two papers, probably because of an insufficient understanding of the molecular mechanism for this gene. We added the name *AVI3* (antiviral RNAi 3) in this work since we found it was induced in leaves by virus infection and conferred virus resistance by enhancing RDR6-dependent amplification of vsRNAs.

The authors state that “Thus, *AVI4* became inactive to interfere with antiviral defense when neither *DCL2* nor *DCL4* was functional.” However, this statement may go a bit too far. Given the very strong effects of deleting *DCL2* and *DCL4* versus the relatively moderate effect of deleting *AVI4*, it seems more likely that any effects of *AVI4* deletion are simply masked by the more important effects of the DCLs.

Response: We agree that deletion of *DCL2* and *DCL4* has very strong effects on virus accumulation. To address this, we have presented a new set of genetic studies in Supplementary Fig. 8 completed since the submission of our manuscript in early September last year. In these studies, we examined the effect of *AVI4* deletion in combination with the deletion of *RDR1* and *RDR6* either alone or together, which has little to less impact than deletion of *DCL2* and *DCL4* on the accumulation of CMV-2aT Δ 2b and CMV- Δ 2b according to our previous studies (ref. 16 and 22). Interestingly, *AVI4* became inactive to significantly enhance the accumulation of CMV- Δ 2b or CMV-2aT Δ 2b in all of the *rdr1* and *rdr6* single or double mutants (Supplementary Fig. 8), suggesting that *AVI4* function requires both *RDR1* and *RDR6*. As described earlier in Fig. 4a-4c, no significant differences were detected in the accumulation of either CMV-2aT Δ 2b or CMV- Δ 2b between *dcl2* and *avi4 dcl2* mutant plants, and in contrast to defense suppression in wild-type plants, *AVI4* appeared to suppress virus accumulation when its main target gene *DCL4* is absent. These findings suggest that *AVI4* suppresses antiviral RNAi only in wild-type Col-0 plants in which the

hierarchical actions of DCLs and RDRs are not perturbed. According to this comment and Reviewer 3's advice, we have added one additional sentence in the Discussion section: "AVI4 is necessary for the transcriptional induction of *DCL2* and the upregulation of *DCL2*-dependent antiviral RNAi by 22-nt vsRNAs in the absence of DCL4, which may explain at least in part the dominant role of DCL4 over DCL2 in the biogenesis of vsRNAs known since 2006". Please see lines 283-290 and 332-335.

The discussion is very brief and there are some issues that could use more discussion including the following:

In the discussion the authors begin with "We show in this work that the natural variation evolved in association with quantitative virus resistance in wild *A. thaliana* populations is under the strongest selection for an optimal antiviral RNAi response. Our findings provide direct evidence to support antiviral RNAi as a dominant defense mechanism to drive host genetic adaptation to viral infection in natural ecosystems." Although this statement is intuitively in line with general thinking, this study does not demonstrate this per se. This study does not demonstrate that natural variation evolved in association with quantitative resistance. In fact, given that these genes are probably involved in myriad other processes (ie. Germination, transcriptional termination) the association could be pleiotropic and should be discussed accordingly. Furthermore, this one study does not demonstrate that RNAi is a dominant defense mechanism. Many other studies have hinted at this in the past.

Response: We have taken this advice and have expanded the discussion section. We have also added in the introduction section: "In this work, we investigated whether single nucleotide polymorphisms (SNPs) significantly associated with virus resistance among wild plant populations are enriched in specific pathways known to confer antiviral protection in plants". Please see lines 45-47.

In the first paragraph of the Discussion, we have added: "In this work, we searched for the natural variation among wild *A. thaliana* populations that is most significantly associated with quantitative resistance to an endemic RNA virus. Surprisingly, none of the SNPs significantly associated with quantitative virus resistance from both of our GWAS screens and an additional GWAS screen reported recently by others mapped to any of the known antiviral RNAi genes, including AGO2 shown recently to exhibit natural variation in non-host virus resistance. However, genetic studies show that both of the highest-ranked gene significantly associated with quantitative virus resistance identified from each of our GWAS screens function in antiviral RNAi". Please see lines 296-307.

Moreover, we also have shortened the abstract as required for journal format by moving several long sentences into the Introduction section and replacing them in the Abstract with "However, it remains unknown whether host single nucleotide polymorphisms significantly associated with virus resistance are enriched in RNA interference (RNAi) pathway genes known to confer essential antiviral defense in plants". Please see lines 13-15.

In addition, we have added in the Discussion: "Whereas *AVI4* has not been characterized before this work, *AVI3* has a known role in seed dormancy with the mechanism yet to be defined. Nonetheless, as the time of seed germination is under strong

natural selection, it is likely that *AVI3* and possibly *AVI4* are also under natural selection for adaptation to other functions in addition to antiviral defense”. Please see lines 322-327.

The authors suggest that *AVI3* and *AVI4* are important factors in determining host-virus interactions. However, the screens have been done with viruses that are extremely compromised in virulence. As such, any conclusions about natural host-virus interactions regarding these genes needs to be tempered in this light.

Response: We have added in the last paragraph of the Introduction section: “we conducted independent GWAS screens with two distantly related CMV strains, one unmodified and the other rendered defective in counter-defense against antiviral RNAi”. We further added in the first paragraph of Discussion: “We conducted independent GWAS screens with wild-type Q-CMV and the VSR-deficient CMV- Δ 2b, respectively. Unlike Fny-CMV, from which CMV- Δ 2b was derived, Q-CMV replicates to higher levels in *dcl2 dcl4* mutant plants than wild-type Col-0 plants (ref. 36, 37), indicating incomplete suppression of antiviral RNAi by Q-CMV”. It should be pointed out that Q-CMV does not replicate to higher levels in *rdr1 rdr6* mutant plants than wild-type Col-0 plants (Supplementary Fig. 4d), suggesting efficient suppression of vsiRNAs amplification by Q-CMV. Strikingly, Fny-CMV does not replicate to higher levels in *rdr1 rdr6* or *dcl2 dcl4* mutant plants than wild-type Col-0 plants (ref. 37), indicating that the antiviral activity of the RNAi pathway cannot be examined in the infection with wild-type viruses that completely suppress RNAi. See lines 58-60 and 296-300.

Reviewer #3 (Remarks to the Author):

This is a nice tidy paper and the presented data mostly support the conclusions that *AVI3* and *AVI4* are regulators of the RDR6 and DCL2 antiviral pathways respectively. The results will be of interest to those in the field, they are an advance on the current literature. Methodology is sound.

I have no substantive comments about the experimental GWAS approach. I do have reservations, however, about the evolutionary interpretation that natural variation in the two genes is due to adaptation of the antiviral response. The two *AVI* genes may well have more general and pleiotropic effects and the authors should not rule out that the variation is associated with adaptive change in these other phenotypes. *AVI3*, for example, is a seed dormancy co-factor. The analysis in this paper does not rule out that the polymorphism in *AVI3* is associated with dormancy or indeed with any other, as yet unidentified, phenotypes of this gene. At the very least the authors should mention the possibility that these genes might be associated with trans-kingdom RNAi associated with fungal and oomycete pathogens even if they cannot test it for this paper.

Response: As advised, we have added in the section of Discussion “Whereas *AVI4* has not been characterized before this work, *AVI3* has a known role in seed dormancy with the mechanism yet to be defined. Our mechanistic studies demonstrate that *AVI3* specifically enhances antiviral RNAi by promoting amplification of the vsiRNAs in a pathway dependent on RDR6, but independent of RDR1. Nonetheless, as the time of seed

germination is under strong natural selection, it is likely that *AVI3* and possibly *AVI4* are also under natural selection for adaptation to other functions in addition to antiviral defense”. Please see lines 323-328.

It would be helpful to have a bit more information about the GWAS alleles of *AVI4*. The interpretation is that the encoded protein is a negative regulator of antiviral RNAi but the GWAS alleles and the *cr5* CRISPR allele all have more quantitative resistance to the CMV isolates – the opposite of what would be expected. If that is the case the GWAS mutations would have affected negative regulatory elements in the *AVI4* promoter so that the encoded protein would be more abundantly expressed in the accessions with the GWAS variant alleles.

Response: We have added experiments and analysis in new panels of Supplementary Figs. 4 and 5 from previously completed experiments that were not included in the first submission. We found that Q-CMV accumulation levels as measured by RT-qPCR were significantly different between the accessions classified as haplotypes A and G according to the identified intergenic SNP (Supplementary Fig. 4c). Northern blot analysis further verified that Q-CMV replicated to lower levels in 3 selected accessions of haplotype G than the 4 accessions of haplotype A including Col-0 (Supplementary Fig. 4d). Moreover, *AVI4* expression levels were significantly higher in the selected accessions of haplotype A than the selected haplotype G accessions after Q-CMV infection (Supplementary Fig. 5b), suggesting enhanced induction of *AVI4* in Col-0 and other haplotype A accessions that support higher Q-CMV replication compared to haplotype G accessions. However, we found no additional SNP within the *AVI4* locus that is significantly associated with Q-CMV titers among natural ecotypes (Fig. 3b). See lines 206-210 and lines 214-216.

One interpretation of *AVI4* is based on the observation that DCLs antagonise DCL21. If *AVI4* promotes this antagonism it would explain why 22nt vsRNA was less abundant in *avi4/dcl4* than in *DCL4* (Fig 4b and d). In tomato at least there are endogenous miRNAs and siRNAs that are dependent on SIDCL2. It would be interesting to know what happens to the Arabidopsis equivalent in *avi4*.

1. Bouché, N., Laressergues, D., Gascioli, V., Vaucheret, H. & Bouche, N. An antagonistic function for Arabidopsis DCL2 in development and a new function for DCL4 in generating viral siRNAs. *EMBO J.* 25, 3347–56 (2006).

Response: We thank the reviewer for this insightful comment on the mechanism by which *AVI4* regulates antiviral RNAi. Accordingly, we have modified our statement in the section of Results as “Consistent with the known dominant role of DCL4 over DCL2 in the biogenesis of vsRNAs. 21-nt vsRNAs made by DCL4 were undetectable whereas 22-nt vsRNAs by DCL2 accumulated to high levels in either *dcl4* plants or *avi4 dcl4* plants (Fig. 4b). By comparison, however, 22-nt vsRNAs accumulated to markedly lower levels in *avi4 dcl4* plants than *dcl4* plants (Fig. 4b), indicating that activation of DCL2-dependent antiviral RNAi by 22-nt vsRNAs in the absence of DCL4 is upregulated by *AVI4*”. See lines 258-263.

We have further added in the section of Discussion: “Interestingly, *AVI4* is necessary for the transcriptional induction of *DCL2* and the upregulation of *DCL2*-dependent

antiviral RNAi by 22-nt vsiRNAs in the absence of DCL4, which may explain at least in part the dominant role of DCL4 over DCL2 in the biogenesis of vsiRNAs known since 2006. However, it is unknown whether *AVI4* also modulates the functional roles of DCL2 that are active in the presence or absence of DCL4 in uninfected plants” (new refs added) . See lines 332-337.

Reviewer #4 (Remarks to the Author):

According to the communications with the editors, I would mainly comment on the GWAS methodologies and the corresponding statistical analysis performed in this study. GWAS was performed by the EMMAX algorithm implemented in the pipeline “easyGWAS”. The EMMAX algorithm is a widely used GWAS algorithm based on the linear mixed model, which is the state-of-the-art method for GWAS in structured populations. Generally, I didn’t detect any major drawback in the statistical analysis of GWAS. But the following points are of my concern:

1) Before GWAS, we would always need to check, if possible, the heritability of the trait (or more precisely, the heritability of the phenotypic values that serve as the input of the GWAS model) because this information tells us the proportion of genetic variation which can be exploited in GWAS. In the Methods section it was mentioned that “all experiments were repeated at least three times” (lines 578-579) and the mapping trait was the mean values of the virus level (measured by ELISA or qPCR, lines 556-558). I would like to ask how large were the correlations between different replicates. Or how was the repeatability of the “phenotypic data”. Strictly speaking, one needs data from several environments/independent experiments to estimate the heritability of the data, which is difficult for some traits like the virus levels inspected in this study. Then, the genomic heritability using linear mixed models may be an alternative. Have the authors considered this?

Response: We have added statements to indicate “viral accumulation measurements in three technical replicates” in the section of “Genome-wide association studies mapping” and “All experiments were biologically repeated at least three times” in the section of “Characterization of virus infection”. See lines 379-380 and lines 427-428.

We have added a sentence “The pairwise Pearson correlations calculated between the three replicates of virus detection were 0.976, 0.970 and 0.972 in CMV-Δ2b screen, and 0.909, 0.905 and 0.925 in Q-CMV screen” (see lines 403-405) as calculated in the Table 1 and Table 2 below. These results indicate a high level of repeatability of these two traits. We also estimated the heritability of ELISA trait and qPCR trait, separately, using the ANOVA approach, which has been stated in the Methods of the revised manuscript. The heritability was 0.9727 for the ELISA trait and 0.9059 for the qPCR trait, which is consistent with the results of the correlation analysis shown in Tables 1 and 2. See lines 401-402.

Table 1. Pairwise correlation between ELISA replicates.

repeat 1	repeat 2	repeat 3
----------	----------	----------

repeat 1	1.000	0.976	0.97
repeat 2	0.976	1.000	0.972
repeat 3	0.97	0.972	1.000

Table 2. Pairwise correlation between qPCR replicates.

	repeat 1	repeat 2	repeat 3
repeat 1	1.000	0.909	0.905
repeat 2	0.909	1.000	0.925
repeat 3	0.905	0.925	1.000

Thanks for suggesting the linear mixed models, which are very useful to estimate environmental effects (fixed effects) and genomic effects (random effects) for calculating heritability. As we stated in the manuscript and the response to this comment, viral accumulation was measured in triplicates under the same condition, so there are no environmental effects. Thus, we leveraged the most commonly used ANOVA method to calculate the trait heritability, which should be simpler than but equivalent to the calculation using the linear mixed model.

2) The authors did not mention how the missing values in the marker data were handled. Did you filter the markers with a certain threshold of missing rate? Did you impute the (remaining) missing values before GWAS?

Response: In the 1001 Genome Project (ref. 9), all the missing genomic data have been imputed using Beagle v3 based on linkage disequilibrium with default parameters, followed by implementing GERMLINE v1.5.1 for an error-tolerant and computationally efficient identification of Identity-by-Descent (IBD) regions for the imputed SNPs. And the pairwise IBD segments were detected as long continuous stretches with a minimum length of 10 kb, merged from slices containing 100 identical SNPs to allow for maximal two mismatches. The easyGWAS pipeline included all of the 1135 natural accessions with 6,973,565 SNPs from the final phase of the 1001 Genomes Project (ref. 26). We believe the 1001 Genome Project has sufficiently handled data imputation, facilitating the use of their genomic data in various secondary analyses. In our study, the minor allele frequency cutoff was set at 5%, yielding 1,823,394 and 1,833,408 SNPs for GWAS mapping of CMV- Δ 2b and Q-CMV screen, respectively.

3) The threshold of P values chosen by the authors could be a problem in the statistical point of view. It was mentioned that a threshold of $1/m$ was applied in GWAS, where m is the number of SNPs used in the analysis (line 558-559). This threshold is far too liberal if one considers the family-wise error rate, for which the threshold is usually $0.05/m$ (the Bonferroni correction). I certainly understand that the Bonferroni correction is in many cases too stringent. But as far as I know, there are three other standard methods implemented in easyGWAS (Benjamini-Hochberg, Benjamini-Yekutieli and Storey-Tibshirani). I am wondering why the authors did not apply these methods. I see that the authors cited a review paper (Zhang et al. 2019) for the choice of the threshold $1/m$. However, this threshold was described as a “subjective and less stringent method” in

the paper, which means that there is no solid statistical theory supporting the choice of this liberal threshold. Of course, experimental verification was done in this study to confirm the function of the candidate genes identified in GWAS, making the choice of threshold a less essential point. Nevertheless, it would be better if one could avoid using subjective thresholds.

Response: We thank the reviewer for mentioning our experimental verification that confirmed the function of the candidate genes identified in GWAS. We wish to point out that in the Q-CMV screen, the peak correlated with *AVI4* was statistically significant using the stringent Bonferroni correction. Nevertheless, the peak associated with *AVI3* was not significant when Bonferroni correction was applied. The similar results were obtained when other adjustment approaches were tried. For example, the $-\log_{10}(\text{q-value})$ of the highest SNP correlated with *AVI3* was 1.27, which is slightly lower than the threshold of 1.30 ($-\log_{10}(\text{q}=0.05)$) as defined by Benjamini-Hochberg and Storey-Tibshirani correction. The results from these procedures largely depend on the number of tests in the multiple comparisons. In our study, about 2 million SNPs (1,823,394 and 1,833,408 for CMV- $\Delta 2b$ and Q-CMV, respectively) were tested, which are almost 10 times larger than many other GWAS studies performed in *A. thaliana* natural accessions with the 250K SNPs chip data (Horton et al., 2012; Rubio *et al.*, 2019; Ogura *et al.*, 2019). We decided to handle our GWAS analysis with 2 million SNPs using the threshold of $1/m$ as this has been adapted by multiple published studies (ref. 67-69). We now included the previous papers of validation of these two genes in the references of the revised manuscript.

1. M. W. Horton, A. M. Hancock, Y. S. Huang, C. Toomajian, S. Atwell, A. Auton, N. W. Mulyati, A. Platt, F. G. Sperone, B. J. Vilhjalmsson, M. Nordborg, J. O. Borevitz, J. Bergelson, 2012. Genome-wide patterns of genetic variation in worldwide *Arabidopsis thaliana* accessions from the RegMap panel. *Nature genetics* **44**, 212-216, 10.1038/ng.1042.
2. B. Rubio, P. Cosson, M. Caballero, F. Revers, J. Bergelson, F. Roux, V. Schurdi-Levraud, 2019. Genome-wide association study reveals new loci involved in *Arabidopsis thaliana* and Turnip mosaic virus (TuMV) interactions in the field. *The New phytologist* **221**, 2026-2038, 10.1111/nph.15507.
3. T. Ogura, C. Goeschl, D. Filiault, M. Mirea, R. Slovak, B. Wolhrab, S. B. Satbhai, W. Busch, 2019. Root system depth in *Arabidopsis* is shaped by EXOCYST70A3 via the dynamic modulation of auxin transport. *Cell* **178**, 400-412, 10.1016/j.cell.2019.06.021.

Reviewers' Comments:

Reviewer #1:

Remarks to the Author:

All the comments and suggestions I did to the previous version of this manuscript have been satisfactorily addressed in the new version, which in my opinion makes a very interesting paper.

Reviewer #2:

Remarks to the Author:

This report essentially describes two different studies aimed at identifying host factors involved in quantitative resistance to viruses in plants. The authors have undertaken two GWAS studies to identify genes displaying natural variation responsible for differing susceptibility to CMV in a large number of Arabidopsis ecotypes and describe the identification of two genes, AVI3 and AVI4.

The AVI3 story is relatively straightforward in that they identify naturally occurring KO alleles and test KO alleles in Col-0 to demonstrate that a lack of functional AVI3 results in increased susceptibility to a VSR-compromised CMV. To understand how AVI3 might function, they generate a number of crosses with lines mutated in RDR and AGO genes. From the results, they conclude that AVI3 enhances vsiRNA amplification. This conclusion is largely based on quantification of virus and vsiRNAs in double/triple mutants. The results from the genetic analyses are consistent with the conclusions. However, without more direct data it is still unclear how AVI3, a putative phosphatase family member, functions in this process.

The second story in this manuscript is the identification of a second locus, AVI4, using a strain of CMV that is naturally less virulent, to undertake the same kind of GWAS analysis. The basis for this gene underlying genetic variation in virus resistance is less clear (see major issue below). Furthermore, the effect on susceptibility is moderate. As such, the same genetic analysis approach used with AVI3 may not be informative because the other RNAi components have so much greater effect than AVI4.

Overall, the data is well executed and the writing is clear. However, there is a lack of converging experimentation to demonstrate that AVI3 and AVI4 are directly involved in RNAi-mediated antiviral defense. AVI3 is putative phosphatase that appears to be involved in germination, whereas AVI4 is annotated as a transcription elongation factor. Indeed, the effects of AVI4 may be due to alterations in the transcription (albeit very moderate) levels of DCL genes rather than being directly involved in RNAi. As such, although these genes are certainly of interest, their direct mechanistic involvement in RNAi remains to be demonstrated.

Major issue:

The authors previously stated that "Thus, AVI4 became inactive to interfere with antiviral defense when neither DCL2 nor DCL4 was functional."

My original concern: Given the very strong effects of deleting DCL2 and DCL4 versus the relatively moderate effect of deleting AVI4, it seems more likely that any effects of AVI4 deletion are simply masked by the more important effects of the DCLs.

The response to this has been to generate more double and triple mutants with RDR1 and 6. The response to the concern is difficult to follow and I simply don't see how this responds to the initial concern. The fact remains that *avi4* has a relatively weak effect on CMV accumulation and thus it is not surprising that it does not substantially affect other more important mutations.

The statement that AVI4 becomes inactive to interfere seems like a poor way of expressing what the authors are implying. Surely the protein does not become inactive in different mutant backgrounds.

The authors state: "AVI4 negatively regulates antiviral defense in Col-0 plants exclusively in the antiviral RNAi pathway"

This statement is unjustified. Furthermore, AVI4 seems to have differing effects in different mutant backgrounds, sometimes apparently inhibiting, sometimes apparently promoting virus accumulation.

AVI4 is a transcription initiation factor and its deletion is expected to have pleiotropic effects. Indeed, the authors' own data suggest that it is required for upregulation of DCLs and thus is very unlikely to have any direct role in RNAi.

Minor issues

Lines 36-38 : "Interestingly, the 19-member AGO family of rice plants includes a hormone-inducible AGO18 to promote antiviral RNAi by enhancing the expression of AGO1 necessary for vsRNA-RISC assembly"

Still grammatical issues. Suggestion:

Interestingly, the 19-member AGO family of rice plants includes a hormone-inducible AGO18 that promotes antiviral RNAi by enhancing the expression of AGO1, which is necessary for vsRNA-RISC assembly

Line 39: "Studies on the origins and variability of viral suppressors of RNAi (VSR) genes have shown that antiviral RNAi exerts selection pressure against plant virus genomes²²⁻²⁶."

Same issue as in the original manuscript. The references cited mostly describe what VSRs are and how they work. Only references 25 and 26 refer to studies pertaining to selection pressure on virus genomes. The response to reviewers mentions new references (27-29). These are not cited in this statement. Regardless, none of these newly referenced studies even mention viruses.

The gene At4g11040 is listed as DOG18 or RDO5 in the Arabidopsis database. Given that a direct effect of virus accumulation is not unambiguous, I still do not see a compelling reason for an additional name.

Reviewer #3:

Remarks to the Author:

The authors have moderated the discussion on lines 327 and 328 in response to the reviewers' comments but the following text remains to be corrected:

line 23 (Our findings reveal a dominant role of antiviral RNAi in driving dynamic host genetic adaptation in natural ecosystems to an endemic viral pathogen

line 63 (Notably, our mechanistic studies demonstrate opposing roles in antiviral RNAi for the two genes evolved in Columbia-0 accession, indicating that antiviral RNAi drives dynamic host genetic adaptation to an endemic viral pathogen in natural ecosystems)

line 340 (In summary, our findings indicate that antiviral RNAi drives dynamic host genetic adaptation to virus infection in natural ecosystems.

The data show an association of virus resistance with genetic variation but they do not show that antiviral RNAi drives dynamic host adaptation to virus infection. In fact, this outcome seems unlikely to me given that the viruses used in this study were disabled or mild strains. A likely scenario, not ruled out, is that the natural variation in AVI3 and AVI4 is driven by other physiological processes and that the virus resistance phenotype is a side effect – pleiotropy.

I would support publication if the discussion points could be further moderated.

The term 'epistatic' on line 189 is not helpful and would be more useful to say that the AVI3 effects

involve the AGO1 but not AGO2 pathway

On line 201 'identified' should be 'identify'

Reviewer #4:

Remarks to the Author:

All my comments and suggestions have been addressed in the revised manuscript.

Point-by-point response to the reviewers' comments

Reviewer #2:

This report essentially describes two different studies aimed at identifying host factors involved in quantitative resistance to viruses in plants. The authors have undertaken two GWAS studies to identify genes displaying natural variation responsible for differing susceptibility to CMV in a large number of *Arabidopsis* ecotypes and describe the identification of two genes, AVI3 and AVI4.

The AVI3 story is relatively straightforward in that they identify naturally occurring KO alleles and test KO alleles in Col-0 to demonstrate that a lack of functional AVI3 results in increased susceptibility to a VSR-compromised CMV. To understand how AVI3 might function, they generate a number of crosses with lines mutated in RDR and AGO genes. From the results, they conclude that AVI3 enhances vsRNA amplification. This conclusion is largely based on quantification of virus and vsRNAs in double/triple mutants. The results from the genetic analyses are consistent with the conclusions. However, without more direct data it is unclear how AVI3, a putative phosphatase family member, functions in this process.

The second story in this manuscript is the identification of a second locus, AVI4, using a strain of CMV that is naturally less virulent, to undertake the same kind of GWAS analysis. The basis for this gene underlying genetic variation in virus resistance is less clear (see major issue below) and the authors do not follow up on this aspect once identifying AVI4. Furthermore, the effect on susceptibility is moderate. As such, the same genetic analysis approach used with AVI3 may not be informative because the other RNAi components have so much greater effect than AVI4.

Overall, the data is well executed and controlled and the writing is clear. However, there is a lack of converging experimentation to demonstrate that AVI3 and AVI4 are directly involved in RNAi-mediated antiviral defense. AVI3 is putative phosphatase that appears to be involved in germination, whereas AVI4 is annotated as a transcription elongation factor. Indeed, the effects of AVI4 may be due to alterations in the transcription (albeit very moderate) levels of DCL genes rather than being directly involved in RNAi. As such, although these genes are certainly of interest, their direct mechanistic involvement in RNAi remains to be elucidated.

Response: We appreciate and agree with these comments. Accordingly, we have made several changes in the revised manuscript (see also below in our response to specific comments). The title of our manuscript is changed to “Natural variation identifies positive and negative regulators of antiviral RNAi defense in *Arabidopsis thaliana*” to emphasize the identification of two genes that display natural variation responsible for differing virus susceptibility and have an indirect role in the regulation of antiviral RNAi. Moreover, AVI3 was changed back to *Reduced Dormancy 5 (RDO5)* named by previous investigators and AVI4 renamed as *Antiviral RNAi Regulator 1 (VIR1)*. We also added the results of one new experiment showing that in contrast to the known phenotypes of *rdo5* mutant plants, *vir1* mutant plants exhibited significantly enhanced seed dormancy (Supplementary Fig. 6e). Thus, our results together indicate that the two genes identified by GWAS in this work have opposing roles in both antiviral RNAi and seed dormancy. We

now mention this in the discussion and propose that a shared mechanism may be under natural selection to regulate antiviral RNAi and seed dormancy.

Major issue:

The authors previously stated that “Thus, AVI4 became inactive to interfere with antiviral defense when neither DCL2 nor DCL4 was functional.” My original concern: Given the very strong effects of deleting DCL2 and DCL4 versus the relatively moderate effect of deleting AVI4, it seems more likely that any effects of AVI4 deletion are simply masked by the more important effects of the DCLs.

The response to this has been to generate more double and triple mutants with RDR1 and 6. The response to the concern is difficult to follow and I simply don't see how this responds to the initial concern. The fact remains that *avi4* has a relatively weak effect on CMV accumulation and thus it is not surprising that it does not substantially affect other more important mutations. The statement that AVI4 becomes inactive to interfere seems like a poor way of expressing what the authors are implying. Surely the protein does not become inactive in different mutant backgrounds.

Response: We agree with the comment. We deleted the statement in the revised manuscript and made a modified conclusion from these experiments “This result indicates that although *VIR1* enhanced antiviral defense in *dcl4* single mutant, it failed to do so in *dcl2 dcl4* double mutant, consistent with the proposed role of *DCL2* in the upregulation of antiviral RNAi by *VIR1*” (see lines 279-282). In addition, we removed the added data from double and triple mutants with RDR1 and RDR6.

The authors state: “AVI4 negatively regulates antiviral defense in Col-0 plants exclusively in the antiviral RNAi pathway”

This statement is unjustified. Furthermore, AVI4 seems to have differing effects in different mutant backgrounds, sometimes apparently inhibiting, sometimes apparently promoting virus accumulation. AVI4 is a transcription initiation factor and its deletion is expected to have pleiotropic effects. Indeed, the authors' own data suggest that it is required for upregulation of DCLs and thus is very unlikely to have any direct role in RNAi.

Response: We agree with the comment and have deleted the word “exclusively” from this sentence and elsewhere in the manuscript. Indeed, the prediction of pleiotropic effects is supported by our new results showing an additional phenotype for this gene in seed dormancy (Supplementary Fig. 6e). Interestingly, a previous study has shown that *rdo2* mutant plants that lack AtTFIIS, which shares homology to AVI4 (but not in the C-terminal half), display clearly reduced seed dormancy. Thus, we have changed the name of this gene from “Antiviral RNAi 4” to “*Antiviral RNAi Regulator 1* (*VIR1*)” to emphasize an indirect role of this gene in the regulation of antiviral RNAi. We also modified the last two sentences in the Discussion section as “In summary, our findings indicate that *RDO5* and *VIR1* have opposing roles in both antiviral RNAi and seed dormancy. We propose that a shared mechanism is under natural selection to regulate

antiviral RNAi and seed dormancy.”

Minor issues

Lines 36-38 : “Interestingly, the 19-member AGO family of rice plants includes a hormone-inducible AGO18 to promote antiviral RNAi by enhancing the expression of AGO1 necessary for vsiRNA-RISC assembly”

Still grammatical issues. Suggestion:

Interestingly, the 19-member AGO family of rice plants includes a hormone-inducible AGO18 that promotes antiviral RNAi by enhancing the expression of AGO1, which is necessary for vsiRNA-RISC assembly.

Response: Modified as recommended – thanks!

Line 39: “Studies on the origins and variability of viral suppressors of RNAi (VSR) genes have shown that antiviral RNAi exerts selection pressure against plant virus genomes22-26.”

Same issue as in the original manuscript. The references cited mostly describe what VSRs are and how they work. Only references 25 and 26 refer to studies pertaining to selection pressure on virus genomes. The response to reviewers mentions new references (27-29). These are not cited in this statement. Regardless, none of these newly referenced studies even mention viruses.

Response: We have modified the references by removing three references and adding two new references associated with selection pressure on virus suppressor of RNAi- thanks.

The gene At4g11040 is listed as DOG18 or RDO5 in the Arabidopsis database. Given that a direct effect of virus accumulation is not unambiguous, I still do not see a compelling reason for an additional name.

Response: As suggested, the original name “Reduced Dormancy 5 (RDO5)” is now used in the revised manuscript for the first gene identified in this work.

Reviewer #3 (Remarks to the Author):

This is a nice tidy paper and the presented data mostly support the conclusions that AVI3 and AVI4 are regulators of the RDR6 and DCL2 antiviral pathways respectively. The results will be of interest to those in the field, they are an advance on the current literature. Methodology is sound.

Response: Many thanks for your positive comments.

The authors have moderated the discussion on lines 327 and 328 in response to the reviewers’ comments but the following text remains to be corrected:

line 23 (Our findings reveal a dominant role of antiviral RNAi in driving dynamic host genetic adaptation in natural ecosystems to an endemic viral pathogen

line 63 (Notably, our mechanistic studies demonstrate opposing roles in antiviral RNAi for the two genes evolved in Columbia-0 accession, indicating that antiviral RNAi drives dynamic host genetic adaptation to an endemic viral pathogen in natural ecosystems)

line 340 (In summary, our findings indicate that antiviral RNAi drives dynamic host genetic adaptation to virus infection in natural ecosystems.

The data show an association of virus resistance with genetic variation but they do not show that antiviral RNAi drives dynamic host adaptation to virus infection. In fact, this outcome seems unlikely to me given that the viruses used in this study were disabled or mild strains. A likely scenario, not ruled out, is that the natural variation in AVI3 and AVI4 is driven by other physiological processes and that the virus resistance phenotype is a side effect – pleiotropy.

I would support publication if the discussion points could be further moderated.

Response: We have modified these sentences as suggested – see lines 24-25, 64-66, and 333-336, and the title of our manuscript is changed to “Natural variation identifies positive and negative regulators of antiviral RNAi defense in *Arabidopsis thaliana*”. We also added the results of one new experiment, which showed that in contrast to the reduced seed dormancy phenotype of the first gene knockout mutant plants, the second gene knockout mutant plants exhibited significantly enhanced seed dormancy (suppl Fig 6e). Thus, our results together indicate that the two genes identified by GWAS in this work have opposing roles in both antiviral RNAi and seed dormancy, which is consistent with the idea that the natural variation in both genes is driven by additional biological functions.

The term ‘epistatic’ on line 189 is not helpful and would be more useful to say that the AVI3 effects involve the AGO1 but not AGO2 pathway.

Response: We have modified this sentence as recommended – thanks.

On line 201 ‘identified’ should be ‘identify’

Response: Fixed – thanks.

Reviewers' Comments:

Reviewer #2:

Remarks to the Author:

The manuscript has been improved by the changes made and my previous concerns have been addressed.

Reviewer #3:

Remarks to the Author:

My concern overlapped with reviewer 2 comments. The concession of pleiotropy is somewhat grudging but the field can decide and I am content that the revised manuscript should be published.

Reviewer #2 (Remarks to the Author):

The manuscript has been improved by the changes made and my previous concerns have been addressed.

Response: Thanks.

Reviewer #3 (Remarks to the Author):

My concern overlapped with reviewer 2 comments. The concession of pleiotropy is somewhat grudging but the field can decide and I am content that the revised manuscript should be published.

Response: Thanks.